# Characterization of Dielectric Oil with a Low-Cost CMOS Imaging Sensor and a New Electric Permittivity Matrix Using the 3D Cell Method

**DOI:** 10.3390/s21217380

**Published:** 2021-11-06

**Authors:** José Miguel Monzón-Verona, Pablo Ignacio González-Domínguez, Santiago García-Alonso, Jenifer Vaswani Reboso

**Affiliations:** 1Electrical Engineering Department (DIE), University of Las Palmas de Gran Canaria, 35017 Las Palmas de Gran Canaria, Spain; josemiguel.monzon@ulpgc.es; 2Institute for Applied Microelectronics, University of Las Palmas de Gran Canaria, 35017 Las Palmas de Gran Canaria, Spain; 3Department of Electronic Engineering and Automatics (DIEA), University of Las Palmas de Gran Canaria, 35017 Las Palmas de Gran Canaria, Spain; santiago.garciaalonso@ulpgc.es; 4Process Engineering Department (DIP), University of Las Palmas de Gran Canaria, 35017 Las Palmas de Gran Canaria, Spain; jenifer.vaswani@ulpgc.es

**Keywords:** cell method, dielectric oil monitoring, imaging sensor, low-cost sensors, permittivity matrix

## Abstract

In this paper, a new method for characterizing the dielectric breakdown voltage of dielectric oils is presented, based on the IEC 60156 international standard. In this standard, the effective value of the dielectric breakdown voltage is obtained, but information is not provided on the distribution of Kelvin forces an instant before the dynamic behavior of the arc begins or the state of the gases that are produced an instant after the moment of appearance of the electric arc in the oil. In this paper, the behavior of the oil before and after the appearance of the electric arc is characterized by combining a low-cost CMOS imaging sensor and a new matrix of electrical permittivity associated with the dielectric oil, using the 3D cell method. In this way, we also predict the electric field before and after the electric rupture. The error compared to the finite element method is less than 0.36%. In addition, a new method is proposed to measure the kinematic viscosity of dielectric oils. Using a low-cost imaging sensor, the distribution of bubbles is measured, together with their diameters and their rates of ascent after the electric arc occurs. This method is verified using ASTM standards and data provided by the oil manufacturer. The results of these tests can be used to prevent incipient failures and evaluate preventive maintenance processes such as transformer oil replacement or recovery.

## 1. Introduction

In electrical power systems, transformers are one of the elements with the highest economic cost, accounting for around 60% of the investment in high-voltage substations. This requires a set of monitoring and diagnostic techniques that affect the life cycle of these important elements. These techniques include the following: dissolved gas analysis, oil quality test, infrared thermography test, power factor test, dielectric dissipation factor test and dielectric oil breakdown test, among others [1,2].

In this paper, an analysis of the quality of the oil is made using a combination of electrical, physical and chemical tests [3]. The transformer oil corresponds to an oil sample without hours of use.

The most important and common tests are related to the dielectric breakdown voltage (BDV), water content, acidity and color. There are three main standards for the determination of the dielectric breakdown voltage of insulating liquids: ASTM D1816–12 (2019), ASTM D877/D877M–19 and IEC 60156:2018. In this paper, UNE-EN 60156, which is based on IEC 60156:2018, was followed.

The results of these tests are used to prevent incipient failures and evaluate preventive maintenance processes, such as transformer oil replacement or recovery [4].

On the one hand, mineral oils in transformers play an important role as an element of electrical insulation between the parts under voltage and, on the other hand, they help to evacuate the heat generated due to hysteresis losses and eddy currents in iron, as well as the losses due to the Joule effect in the transformer coils. This last condition requires the oil to have a high thermal conductivity and a low coefficient of dynamic viscosity.

The breakdown strength of dielectric oils for transformers will depend on the nature of the impurities present in their solid or gaseous state. Oil analysis is important for extending the life of the transformer.

The state of knowledge of dielectric breakdown voltages in insulating liquids is less developed than in the case of gas and solid dielectrics. The studies carried out may in some cases be contradictory [5].

Among these studies are those that explain the dielectric breakdown voltages of liquids based on an extension of the dielectric breakdown voltages in gases, in turn based on the avalanche ionization of the atoms caused by electron collisions in the applied field [6].

Dielectric breakdown voltages in different temperature ranges show little dependence on temperature. This suggests that the cathode emission process is one of field emission rather than thermionic emission [6].

Electronic theory predicts the relative magnitudes of the dielectric breakdown voltages well, but not the times at which these breakdowns occur in the insulating liquid. This last aspect—the temporal aspect—is partly explained by the presence of polluting particles inside the insulation. These give rise to local breakdowns, which in turn lead to the formation of small bubbles with a much lower dielectric strength and, hence, finally lead to breakdown.

Other phenomena that explain electrical breakdown include the electroconvection of dielectric breakdown, dielectric liquids subjected to high voltages and electrical conduction resulting mainly from charge carriers injected into the liquid from the electrode surface. The resulting space charge gives rise to Coulomb’s force which, under certain conditions, causes hydrodynamic instability, creating an eddy motion of the liquid which yields a convection current.

Thus, the charge transport will be largely via liquid motion and not ionic drift. The key condition for the instability onset is that the local low velocity exceeds the ionic drift velocity [5].

In most of the studies referenced in the bibliography dealing with analyses of the dielectric breakdown voltage in dielectric oils, approximate analytical equations are used, or numerical methods are employed based on a differential formulation such as the finite elements method (FEM).

In the present study, we propose the finite formulation (FF) method [7], together with the cell method (CM) [8,9] as an associated numerical method to analyze this type of device. Using this methodology, we consider the global magnitudes associated with space-oriented elements such as volumes, surfaces, lines and points of the discretized space, as well as temporal elements, instead of field magnitudes associated with independent variables, i.e., spatial and temporal coordinates [7].

In addition, equations of the constitutive type—equations of the medium—are clearly differentiated from those of the topological type—equations of balance. In FF, the physical laws that govern the electromagnetic equations are expressed in their integral form. In this way, the final system of equations is obtained directly, without the need to discretize the equivalent differential equations [8].

In the CM, the topological equations obtained directly from Maxwell’s laws are exact (balance equations), while the constitutive equations obtained from the discretization process are approximate. In the latter case, source-type quantities defined in the elements of the dual mesh must be related to the configuration quantities corresponding to the elements of the primal mesh [10]. Field magnitudes and the physical properties of the medium are assumed to be constant, at least in the primal mesh. This ensures that the discrete equations are consistent with the continuous constitutive equations, in the sense that the discrete constitutive equations approximate the continuous constitutive equations with an error that decreases with the mesh size [11].

Most of the research papers in the CM literature focus on the construction of discrete constitutive equations. Among those related to a quasi-electrostatic problem in 2D with plane symmetry, is [12]. In this paper, an isotropic and anisotropic electrostatic field is studied by means of the CM. In [13], the electrostatic problem is studied in 2D with plane symmetry. The constitutive equation is used with two approximations. The first approach assumes a uniform field with a triangular base inside each primal cell, and the second approach, which is more general, assumes the uniformity of the fields in subregions of each primal cell, with quadrilateral bases. In [14], a 2D analysis with axial symmetry (axisymmetric) is performed for a quasi-electrostatic problem regarding a gas-insulated line for an ITER neutral beam injector.

In [15], an electrostatic induction micro-motor is studied, using the CM in 2D.

In the literature, to the best of the authors’ knowledge, the 3D cell method has not hitherto been used to simulate dielectric breakdown voltage tests on transformer oils as a quasi-electrostatic problem. In this article, we propose the use of a geometric structure for the electrical permittivity constitutive matrix, analogous to the matrix that appears in [16] for an electromagnetic problem in 3D to calculate eddy currents.

The advantage of using the same geometric structure in the constitutive matrices of conductivity and permittivity in the quasi-electrostatic problem in 3D is that it reduces the complexity of the programmed source code and the execution times. This is because the constitutive matrix is calculated in the assembly of the system of equations, only once for each tetrahedron. The electrical conductivity and electrical permittivity properties of each tetrahedron are multiplied by the common matrix. This is done, element by element, until the final system of equations is complete.

In the present study, a new constitutive matrix is formulated that, using the CM, relates the differences in electrical potentials (magnitudes of configuration) due to primal mesh edges with the electrical flux (magnitudes of source) due to dual mesh planes. The magnitudes of configuration are associated with the edges of a primal mesh made up of tetrahedra, and the source-type magnitudes are associated with the surfaces of a dual mesh (the control volume) obtained in a barycentric division of the primal mesh.

This paper presents an experimental study of the dielectric strength of transformer oil based on the IEC 60,156 standard [17]. Our contribution consists of characterizing the behavior of the oil an instant before and after the electric arc rupture, combining a low-cost complementary metal oxide semiconductor (CMOS) imaging sensor and a new electrical permittivity matrix $M_{\varepsilon }$, using the 3D CM [7,15]. In the standard test, only the effective value of the dielectric breakdown voltage is obtained. However, the information on the distribution of Kelvin forces [18] an instant before the dynamic behavior of the arc begins is lost, as is the information on the gases that are produced an instant after the moment of breakdown via the electric arc in the oil.

This last aspect was analyzed by recording images of the movement of the gas bubbles that are produced within the oil. This also allowed the diameter of these bubbles to be measured. The measurement of their magnitudes was used to indirectly obtain the viscosity of the oil. The physical property of viscosity could be obtained by analyzing the post-arc images using an equation to predict the terminal velocity of the rise of isolated bubbles in Newtonian liquids [19].

The data obtained with the sensors and the results of the simulations complement each other and offer information that would otherwise be lost when strictly following the standard test.

The use of low-cost camera systems in remote-sensing applications is not new. The use and study of low-cost cameras for engineering and scientific applications was addressed in detail in [20]. In [21] a study of the corona effect in aeronautical applications was performed, using low-cost Raspberry-Pi-type cameras. In [22], an array of single-board computers produced by Raspberry Pi, and their associated 8 MP cameras, was used at the University of Cambridge to capture the images required for particle image velocimetry analysis or analysis of the correlation of digital images.

There are two main objectives in this paper. One is to present a new matrix of electrical permittivity, $M_{\epsilon }$, that predicts the electric field before and after the electric rupture occurs. The other is to measure the kinematic viscosity of the dielectric oil using a low-cost CMOS imaging sensor to measure the distribution of bubbles, their diameters and their rates of ascent after the electric arc occurs. In addition, experiments were performed to estimate the dielectric breakdown voltage in order to obtain the boundary conditions for CM and FEM simulations. In this way, both objectives are related.

This paper has been divided into the following sections. Section 2 determines the distribution of $\vec{E}$ and $\nabla E^{2}$ in the dielectric strength test by applying the 3D CM using the new constitutive matrix $M_{\epsilon }$. Section 3 describes the low-cost 8 MP CMOS imaging sensor used in the experimental studies. Section 4 presents in detail the numerical results of the CM simulations with $M_{\epsilon }$. This matrix is verified by comparing the results obtained with those from the FEM analysis. Finally, Section 5 presents the experimental setup of the oil testing device and the results that were obtained. In addition, it lays out the test procedure for the dielectric strength and kinematic viscosity of the transformer oil and establishes the theoretical basis of the new procedure for the determination of the dynamic viscosity of the oil. Finally, the data obtained on the kinematic viscosity using the proposed methods are verified by comparing them with the manufacturer’s data.

## 2. Distribution of $\vec{E}$ and $\nabla E^{2}$ in the Dielectric Strength Test

The formation of the electric arc and the subsequent bubble formation are highly dependent on the estimation of the electric field distribution. Furthermore, its gradient determines the forces per unit volume that act on polluting particles and microbubbles within a dielectric [23,24].

It is important to determine the distribution of the field $\vec{E}$ and the gradient of its square $\nabla E^{2}$ as these are factors that determine the DBV.

The following section explains the proposed method for obtaining this field distribution using the 3D CM as an alternative method to the FEM. In this section, a new electric permittivity matrix $M_{\epsilon }$ is proposed for use in the CM.

### 2.1. New Constitutive Matrix $M_{\varepsilon }$. Discrete Constitutive Equations of Transformer Oil in the Finite Formulation

The electrical constitutive equation in transformer oils is a complex equation based on the Fowler–Nordheim theory [25]. In most dielectric materials, the conduction current of free carriers is relatively low, since their conductivity is usually several orders of magnitude lower than that of a metal or semiconductor. In new transformer oils at 50 °C it is usually of the order of 1 × 10^−13^ S/m, and in used oils of the order of 1 × 10^−11^ S/m [26].

In this paper, a conductive-type model is considered and the conductivity is assumed to be the same throughout the volume of the oil, where the volumetric current density $\vec{J}$ is directly proportional to the electric field $\vec{E}$ [25].

Taking into account the non-zero conductive properties of oil and the fact that it is subjected to an electric field, in the CM the current through the material can be described by its constitutive equation of current flow as a function of potential differences associated with the edges of the primal mesh *e_i_ i =* 1:6, as shown in Figure 1. The constitutive equation shown in Equation (1) was developed in [16].
(1)$${\tilde{I}}_{f}=M_{\sigma }\cdot U.$$

The electrical constitutive matrix $M_{\sigma }$ is given by the expression $M_{\sigma }=\frac{\sigma }{\nu }{\tilde{S}}_{i}\cdot {\tilde{S}}_{j}\;i;\;j=1:6$. The matrix $M_{\sigma }$ is a function of the electrical conductivity of each tetrahedron; its volume and the dot product of the surface vectors correspond to the dual planes (green planes) of primal edges, edges *e_i_* and *e_j_ i j* = 1:6, as shown in Figure 1.

In dielectrics, however, when the applied electric fields are variable with time, a new contribution to the free current appears, the so-called displacement current. This appears when there is variation in the electric flux with respect to time. The displacement current has two terms: ${\tilde{I}}_{d}=\frac{\partial \tilde{\psi }}{\partial t}=\frac{\partial \left({\tilde{\psi }}_{0}+{\tilde{\psi }}_{P}\right)}{\partial t}$. The first term within the parentheses only depends on the potential difference in a vacuum. It is independent of the characteristics of the material. The second term depends on the insulating material used. This depends exclusively on the polarization of the dielectric—in this case the transformer oil—and contains the response of the material, which will be different according to the polarization mechanisms that occur for each stimulus of the net applied potential difference (due both to free charges and to those of polarization). We can group both terms with the total empty and material medium permittivities in $\varepsilon =\varepsilon_{r}\varepsilon_{0}$, with the constitutive equation given in Equation (2).
(2)$$\tilde{\psi }=M_{\varepsilon 0}U+M_{\chi P}U=M_{\varepsilon }U.$$

Equation (2) relates the differences in electric potential U associated with the edges of the primal mesh (magnitudes of configuration) with the electric fluxes $\tilde{\psi }$ of the dual planes ${\tilde{S}}_{i}\;i=1:6$, as shown in Figure 1. In Equation (2), the matrix $M_{\varepsilon }$ is the new electrical constitutive matrix proposed in this work. Given the analogy with Equation (1), the following expression is proposed:(3)$$M_{\varepsilon }=\frac{\varepsilon }{\nu }{\tilde{S}}_{i}\cdot {\tilde{S}}_{j}\;i;\;j=1:6.$$
where $M_{\varepsilon }$ is a function of the electric permittivity of each tetrahedron, $\varepsilon =\varepsilon_{0}\varepsilon_{r}$, its volume ν and the dot product of the surface vectors that correspond to the dual planes of primal edges, edges *e_i_* and *e_j_*. The value of $\varepsilon_{r}$ depends on the type of oil. At 20 °C, this lies between the values of 2.1 and 3.5 [27]. Therefore, the displacement current is obtained from Equation (4).
(4)$$\;{\tilde{I}}_{d}=\frac{\partial \left(M_{\varepsilon }U\right)}{\partial t}.$$

The total current will be given by the sum of both contributions, according to Equation (5). The total current ${\tilde{I}}_{t}$ is represented for the dual plane ${\tilde{s}}_{3}$ in Figure 1.
(5)$${\tilde{I}}_{t}={\tilde{I}}_{f}+{\tilde{I}}_{d}.$$

### 2.2. Maxwell’s Laws in Finite Formulation Applied to Transformer Oil

Maxwell’s laws, applied in their finite formulation in the dielectric strength test are, first of all, the laws corresponding to the configuration magnitudes. According to [8], these are as follows.

(a)Gauss’s law for the magnetic field, Equation (6)
(6)D(Φ)=0,
where *D* is the volume–face incidence matrix of the primal mesh, which is equivalent to the standard divergence operator. The magnitude ϕ represents a vector with all the magnetic fluxes associated with the four faces of the primal mesh tetrahedron if *i* = 1:4, as shown in Figure 1.(b)Faraday’s law of induction, Equation (7)
(7)$$C\cdot U=-\frac{\partial \left(\phi \right)}{\partial t},$$
where *C* is the face–edge incidence matrix of the primal mesh, which is equivalent to the standard rotational operator. U is a vector of potential differences extended to all edges of the primal mesh and *t* is time.

The next two laws correspond to the laws that operate with magnitudes of energy.

(c)Generalized Ampere’s law, Equation (8)
(8)$$\tilde{C}\cdot \tilde{F}={\tilde{I}}_{f}+\frac{\partial \left(\tilde{\psi }\right)}{\partial t},$$
where $\tilde{C}$ is the face–edge incidence matrix in the dual mesh, the vector $\tilde{F}$ is a vector of magnetomotive force associated with all the edges of the dual mesh, ${\tilde{I}}_{f}$ is a vector of electric currents extended to all planes of the dual mesh and, finally, $\tilde{\psi }$ is the electric flux due to the polarization of the dielectric associated with the faces of the dual mesh.(d)Gauss’s law of the electric field, Equation (9)
(9)$$\tilde{D}\cdot \tilde{\psi }=Q_{f},$$
where $\tilde{D}$ is the incidence matrix of the volumes–faces of the dual mesh and $Q_{f}$ is the charge contained in each dual volume.

Finally, if the divergence is applied to Equation (8), taking into account Equation (9), Equation (10) is obtained.
(10)$$\tilde{D}\cdot {\tilde{I}}_{f}+\frac{\partial \left(Q_{f}\right)}{\partial t}=0.$$

### 2.3. Maxwell’ Laws and Constitutive Equations

In this section, the constitutive equations and Maxwell’s laws given in Section 2.1 and Section 2.2 are combined to obtain the final equation in the time and frequency domain. The electrical scalar potential is used for this purpose, and it significantly reduces the number of unknowns.

As we mentioned in Section 2.1, we assume that the electrical conductivity of the oil is low, of the order of magnitude of $\sigma =1\times 10^{-13}$ S/m. The permittivity of oil is $\;\varepsilon =\varepsilon_{r}\cdot 8.854187818\times 10^{-12}$ F/m. Three time constants can be considered to fit the type of problem to be solved. The first is defined as the charge-relaxation time, τe=εσ≈εr⋅88 s. The second is the electromagnetic time constant, τem=lc≈10−12 s, where l = 10 cm, the characteristic length of the domain. The constant c is the speed of light. The third constant is the magnetic time constant, τm=τem2τe≈10−22 s. We can consider that if the frequency is 50 Hz, τ ≈ 20 ms and the conditions for the field to be quasi-electrostatic are β=(τemτ)2≪1 and $\tau_{m}<\tau_{em}<\tau_{e}$. For further information on quasi-static laws and time rate expansions, see [28]. In this way, the field can be considered to be quasi-electrostatic, and Equation (7) can be written as
(11)C·U≈0

Since the field is, in this situation, almost electrostatic, it is possible to work with a single electric potential U=−Gφ, where φ is an electric scalar potential. This is imposed on the surface of the electrodes, as shown in Figure 2. Taking into account the constitutive Equations (2) and (9), this is as follows.
(12)$$D\cdot M_{\varepsilon }\left(-G\varphi \right)=Q_{f}$$

Taking into account Equation (10), and substituting in this equation the free volumetric electric charge *Q_f_* from Equation (12) and the free current $I_{f}$ from Equation (1), using U=−Gφ, Equation (13) is obtained.
(13)$$\tilde{D}\cdot M_{\sigma }G\varphi +\frac{\partial \left(\tilde{D}\cdot M_{\varepsilon }G\varphi \right)}{\partial t}=0$$

Equation (13) corresponds to the differential equation derived in [28]. If the electrodes work at a frequency of *f* = 50 Hz, with an angular frequency of *ω* = 2π50 s^−1^, the final equation in the frequency domain will be Equation (14). It can be observed that this equation involves electrical permittivity and electrical conductivity. This is the equation to be programmed, together with the global electrode current ${\tilde{I}}_{t}$, which is calculated using Equation (15) where $I_{c}$ is an incidence vector of the relative cut between the edges of the oil volume mesh and the surface of one of the electrodes, as shown in Figure 3. The sum of all currents in that cut is equal to the total current entering or leaving at each of the electrodes.
(14)$$\tilde{D}\cdot M_{\sigma }G\varphi +j\omega \tilde{D}M_{\varepsilon }G\varphi =0$$
(15)$${\tilde{I}}_{t}=-I_{c}\cdot \left(M_{\sigma }G+j\omega \tilde{D}M_{\varepsilon }G\right)\varphi$$

The matrix representation of both equations is shown in Equation (16).
(16)$$\left[\begin{array}{cc} G^{t}M_{\sigma }G+j\omega G^{t}M_{\varepsilon }G & 0\\ I_{c}M_{\sigma }G+j\omega I_{c}M_{\varepsilon }G & 1 \end{array}\right]\left[\begin{array}{c} \varphi \\ {\tilde{I}}_{t} \end{array}\right]=\left[\begin{array}{c} 0\\ 0 \end{array}\right]$$

The unknowns are all the potentials φ of the primal mesh nodes and the global magnitude of the current ${\tilde{I}}_{t}$ associated with the surface of one of the electrodes.

Note that the total current has been defined twice by Equations (5) and (15) because Equation (5) shows the two main components of the total current, i.e., the displacement current and the conductive electric current, while Equation (15) details its explicit composition based on its physical and geometric properties.

In the matrix system (16), the first row is uncoupled from the second. However, we have preferred a compact equation system that includes the total intensity of the electrode. This avoids an additional post-processing calculus. It is also true that we have increased the dimensions of the system by one degree of freedom, corresponding to the total intensity through the electrode. In this way, by solving a single matrix system all the unknowns (degrees of freedom) are obtained at the same time, without post-processing.

Boundary conditions are stabilized on the electrode surfaces. All the nodes on one electrode surface have a value of the electric potential equal to zero and all the nodes on the surface of the other electrode have the dielectric breakdown voltage.

The electrodes have an axisymmetric geometry, but the oil container is cubic, making the global model non-axisymmetric. Hence, we must perform a 3D analysis. Furthermore, the new proposed electrical permittivity matrix is a 3D matrix, and these calculations give generality to the matrix $M_{\varepsilon }$.

### 2.4. Kelvin Polarization Forces in Dielectric Materials

The force suffered by a supposed spherical particle or a micro-bubble of gas in suspension of radius *r* and relative permittivity ε, in a liquid with relative permittivity $\varepsilon_{oil}$, in the presence of an electric field $\vec{E}=\left(E_{x},E_{y},E_{z}\right)$, is calculated using the Kelvin polarization force formula, according to [23], as shown in Equation (17).
(17)$${\vec{F}}_{e}=\overset{r}{\underset{0}{\int }}\epsilon_{0}\left(\epsilon -\epsilon_{oil}\right)\vec{E}\cdot \nabla \vec{E}\;dv.$$

If the x component of ${\vec{F}}_{e}$ is developed, Equation (18) is obtained.
(18)$$F_{e}^{x}=\varepsilon_{0}\left(\varepsilon -\varepsilon_{oil}\right)\left(E_{x}\frac{\partial E_{x}}{\partial x}+E_{y}\frac{\partial E_{y}}{\partial y}+E_{z}\frac{\partial E_{z}}{\partial z}\right).$$

It is developed in a similar way for *y* coordinates as for *z* coordinates. As $\nabla \times \vec{E}\approx 0$, Equation (17), according to [24], is simplified and Equation (19) is obtained.
(19)$${\vec{F}}_{e}=\overset{r}{\underset{0}{\int }}\varepsilon_{0}\left(\varepsilon -\varepsilon_{oil}\right)\frac{1}{2}\nabla E^{2}\;dv.$$

If $\varepsilon >\varepsilon_{oil}$, this force tends to move the particles to the area where the field is stronger, aligning them and forming a bridge that makes it easier for the current to cross the liquid dielectric via that path. The field in the area of the particles increases and the breakdown value is reached [5].

If the number of particles is not sufficient to bridge the gap, the particles give rise to a local field enhancement, and if the field exceeds the dielectric strength of the liquid, local breakdown will occur near the particles, thus resulting in the formation of gas bubbles which have a much lower dielectric strength and hence, finally, leading to breakdown.

It is important to highlight the dependence of the electric field strength and its gradient and hence the importance of the estimation of the distribution of this electric field and its gradient. This is achieved by applying the method proposed in this paper: the CM together with the proposed new electrical permittivity matrix $M_{\varepsilon }$.

## 3. The Low-Cost 8 MP CMOS Imaging Sensor

### 3.1. Camera Features

Sony IMX219 cameras, version v2.1, with 8 megapixels and CMOS sensors, were used in all the experiments. They can be seen in Figure 4, and a detail of the central camera is shown in Figure 5. The plates of the camera weigh 3 g and measure 25 × 24 mm. They are much lighter than most other computer vision cameras and significantly cheaper, costing around USD 30. Each camera requires its own Raspberry Pi to control all its parameters. The small size of the camera and its various elements, make it easy to assemble at a low cost and to connect to its paired computer, compared with other more traditional computer vision cameras or digital cameras.

The camera, as a sensor, must be perfectly linear throughout the detection area. The ideal sensor produces the same response to a photon striking the surface in the center as it does to one at the edge of the detection area. In [29], the data plotted for the Raspberry Pi camera showed that, for the standard camera operating mode, the gain did not allow radiance values higher than 0.4 $\frac{W}{m^{2}sr}$, but the signal produced at these lower light intensities was very linear. Cameras without a near-infrared (NIR) filter have different gains with similar compensations [29].

The cameras can capture fixed photos at a resolution of 3280 × 2464 pixels or high-definition video at a resolution of 1920 × 1080 pixels and a rate of 30 frames per second (fps). Higher frame rates have been achieved via custom manipulation of the camera interface using the OpenCV software library [30], with 640 × 480 resolution and capture rates of up to 87 fps obtained in all the experiments carried out.

### 3.2. Camera Triggering with Raspberry Pi

A Raspberry Pi is a small board of a size similar to a credit card which has an ARM microprocessor with power of up to 1 GHz, integrated in a Broadcom BCM2835 chip. One of the features of the Raspberry Pi is the electronic general-purpose input/output (GPIO) interface [22].

This is a built-in 40-way connection that can be used to send and receive 3.3 VDC signals, as shown in Figure 6. It was used in the configuration of the cameras described in this paper to activate three cameras simultaneously. They were controlled through an interface programmed in Python, in the Raspberry Pi operating system. The operating system used by the Raspberry Pi is called Raspbian and it is a distribution of the GNU/Linux operating system based on Debian and therefore free to use. Synchronization of all the cameras was achieved by electronically and simultaneously activating the cameras through a switch that triggered them through the GPIO inputs on each Raspberry Pi. These were connected to each other and to a central computer to collect the data, as shown in Figure 4.

The camera–Raspberry Pi system is ideal for performing image analysis in near real time, using the numerical analysis libraries in Python, such as the fundamental package for scientific computing with Python, NumPy [31]. There is also extensive support for the OpenCV computer vision package.

A method of interacting with the Raspberry Pi camera board through a Python interface via the picamera software library was used [32]. This high-level software provides the user with a library of Python commands to manipulate the GPU on the Raspberry Pi and control the camera settings through software manipulation of the data captured by the CMOS image sensor. The Raspberry Pi computers are configured by a startup program that runs automatically on each of the three Raspberry Pi computers paired with their corresponding cameras, i.e., three Raspberry Pi computers and three cameras, as shown in Figure 4.

## 4. Numerical Results of the Simulations in CM with $M_{\varepsilon }$ vs. FEM

The system of equations shown in Equation (16), in a sinusoidal steady state, was programmed in C++. As the matrices are sparse and of large dimensions, the Krylov subspace was used. These algorithms were implemented in the PETSc software [33].

In particular, the linear solver employed was the generalized minimal residual algorithm (GMRES).

The main characteristics of the computer used to make the simulations were: computer model: X399 AORUS PRO; architecture: x86_64; total memory: 128 GB; processors: 24; cpu: 2185.498 MHz; thread(s) per core: 2; core(s) per socket: 12.

The numerical validation of the proposed constitutive equation as shown in Equation (2) and the system of equations shown in Equation (16) was carried out by comparing it with the standard FEM and its implementation within the GetDP software [34]. A very dense reference mesh was used in 3D with a number of tetrahedra equal to 2,790,589 volumes and 487,435 nodes. The Gmsh software [35] was used for the mesh and the data visualization. The nodes determine the number of unknowns in the system of equations, as shown in Equation (16).

A section of this mesh and the solution of the potential distribution corresponding to a potential difference between the electrodes of 19,488 V for the dielectric breakdown voltage, can be seen in Figure 7.

The results corresponding to the spatial distribution of the densities of the conductive and displacement currents are shown in Figure 8 and Figure 9, respectively.

The results of the distribution of the force density per unit volume scaled with the coefficient $\frac{1}{2}\left(\varepsilon -\varepsilon_{oil}\right)$ are shown in Figure 10 and Figure 11. These graphs were obtained by cutting the simulation result with the planes 1 X + 0 Y +0 Z − 0.0489 = 0 and 0 X + 1 Y + 0 Z − 0.035 = 0, respectively. Note that the maximum force occurs in a ring around the center.

Figure 12 shows the capture of the electric arc recorded by cameras A, B and C, from the left, center and right, respectively. The images show the breakdown point of the arc in an area that coincides with the one estimated in the model of Equation (19).

Furthermore, the usefulness of calculating Kelvin forces an instant before the dielectric breakdown voltage is reached, using CM, helps in understanding where the dielectric breakdown voltage occurs. Figure 10 and Figure 11 show that maximum forces are located in a ring around the electrode, but not at its center. This is coherent with the experimental images shown in Figure 12, obtained with the imaging sensors.

### 4.1. Validation of the Numerical Simulations

This section presents the numerical experiments to validate the results obtained by the proposed method, i.e., the cell method CM and the permittivity matrix $M_{\varepsilon }$ from Equation (3). This is compared with the FEM in 3D, with a high number of elements (tetrahedra). The latter adapt well to the surface of the electrodes. The geometry of the problem has planes of symmetry. In the end, this always becomes a 3D problem. The experiments solved the entire problem in 3D.

Table 1 summarizes the geometric and physical properties of the dielectric oil used in the numerical experiments. Each type of experiment E1, E2 and E3 was subdivided into two data analyses corresponding to cut A and cut B, which are two characteristic areas of the electrodes, as shown in Figure 7. Between these two zones, the potential gradients differ greatly and serve to compare the two numerical methods of CM and FEM–getdp using the same mesh.

The number of points in the cut used in all the experiments was 180. The analysis of the metrics was carried out on this basis.

Three numerical experiments were designed: E1, E2 and E3. These consisted of discretizing the electrodes and the volume around them, as shown in Figure 7. Each experiment had a different mesh density, as indicated in Table 1. The objective of these experiments was to check the convergence of the CM numerical method, as well as to determine the error produced when comparing it with another numerical reference method, FEM–getdp.

#### 4.1.1. Results of Experiment E1

Figure 13 shows the results of the simulation using the CM and the FEM–getdp. These are the distributions of electric potential obtained in cuts A and B. The lines with a higher slope correspond to cut A and those with a lower slope to cut B.

Figure 14 shows the distribution of the electric field strength modules. A greater electric field is observed for cut A.

In experiment E1, a low-density mesh was used. In Figure 13, it can be seen that the voltage values in the FEM–getdp and CM are almost coincident for cut A. However, they differ more for cut B. In Figure 14 there is a notable difference between the values of the electric field given by the FEM–getdp and those given by the CM, in both cut A and cut B, because the mesh is not very dense.

#### 4.1.2. Results of Experiment E2

Figure 15 and Figure 16 represent the results for the electric potential and the modules of the electric field strength, respectively, corresponding to experiment E2.

In experiment E2, a medium-density mesh was used. In Figure 15, it can be seen that the voltage values for the FEM–getdp and CM are practically the same. In Figure 16, there are still differences between the electric field values given by the FEM–getdp and CM, in both cut A and cut B.

#### 4.1.3. Results of Experiment E3

The maximum convergence corresponds to Figure 17 and Figure 18, with a total number of tetrahedra of 2,790,589 for experiment E3, as shown in Table 1.

A high-density mesh was used in experiment E3. In Figure 17 and Figure 18, there are no differences in the values given by the FEM–getdp and CM for either the voltage or the electric field.

Therefore, it is observed that as the mesh density is increased, the two methods give coincident solutions. This confirms the validity of the new *M_ε_* matrix proposed in this paper for the CM.

### 4.2. Metrics of Numerical Experiments

To validate the proposed method ($M_{\varepsilon }$ using the CM), a series of comparisons was established between the results obtained with the CM and with the FEM–getdp in the numerical experiments performed.

Metrics are applied to the validation of a model against a reference or pattern. In this case, the model to be validated was the results obtained with the CM and the reference or pattern was the results obtained with the FEM. The CM and FEM are approximate numerical methods, therefore not exact. In both methods a tolerable error is pre-set.

There are various sources of error. One is the truncation of the figures and the accumulation of errors due to the numerical operations performed. Another, which is the one that most affects our problem, is the layout of the cuts. The proximity of the cut to the nodes of the mesh makes the calculated value at the cut more accurate. In contrast, when the cut moves further away from the node, it will be necessary to obtain interpolated values, thus producing a greater error. This happens in both the FEM and CM.

In this study, an analysis was carried out in the FEM with a very dense mesh. These results were used as a reference. Different mesh densities were established in the CM. With the results obtained using the CM, different comparisons were made with the results of the high-mesh-density FEM model. Various statistical indicators (metrics) were used to check the validity of the CM versus the FEM.

The comparisons made are shown in Table 2.

Of the various statistics that can be used to measure the goodness of fit of a model, the following were chosen for the present study: the coefficient of determination (R^2^), the root mean square percentage error (RMSPE), the mean absolute percentage error (MAPE) and the percentage bias (PBIAS). Table 3 and Table 4 show the range of these statistics, their optimal values and the value obtained in each the comparisons carried out. When the value obtained is closer to the optimum of the statistic, the analyzed mathematical model has a better goodness of fit.

Figure 19 shows various error histograms for some of the comparisons carried out. Following error theory, an ideally random error distribution should have its mean around zero and a Gaussian or normal distribution.

If the comparisons are grouped following the order {C1, C5, C9}, {C2, C6, C10}, {C3, C7, C11} and {C4, C8, C12}, it is observed, in a generalized way, that increasing the mesh density improves the model for any of the numerical experiments analyzed.

Comparing Figure 13, Figure 15 and Figure 17 with Figure 14, Figure 16 and Figure 18, it can also be seen that the voltages in {C3, C7, C11} and {C4, C8, C12} were modeled, in relative terms, much better than the electric fields in {C1, C5, C9} and {C2, C6, C10}.

The values of the coefficient of determination (R^2^) indicate a good fit of the data for all the comparisons except those cases where the mesh was not very dense. The RMSPE, MAPE and PBIAS values are relatively high in the comparison. Nevertheless, all the indicators are in the optimal range. Even the largest error value of 0.36% for the RMPSE, as shown in Table 3, is a more than acceptable value.

The distribution of the errors, following error theory, conforms to a normal distribution centered on the zero value, as shown in Figure 19.

The detailed formulations of the metrics used appear in the Annex.

## 5. Experimental Setup of Oil Testing Device

### 5.1. Procedure for Dielectric Strength Test and Kinematic Viscosity of Oil

Dielectric oil strength test equipment [17] is known as a spark meter or dielectric oil tester. The equipment is in the form of a closed metal cabinet with a handle to facilitate its transport, as shown in Figure 20. Inside is the test cell, where the oil sample to be tested is placed. The tester is equipped with detachable electrodes for testing with different types of electrodes. The cell has a magnetic stirrer in the lower part of the oil deposit for perfect homogenization of the oil sample to be tested, and the rotating part is a small Teflon magnet, as shown in Figure 20.

The technical characteristics of the equipment are summarized in Table 5.

### 5.2. Performed Experiments

In order to obtain data prior to using the proposed method, a series of standard tests were performed. All the tests carried out on the transformer oil sample are shown in Table 6, together with the test methodology [39,40,41,42,43,44].

In view of the obtained results, it can be confirmed that the parameters analyzed in the oil used complied with the specifications recommended by the UNE 60296:2012 standard [44].

In addition to the previous tests, a set of additional experiments were carried out with the objective of carrying out a strength test and estimating the viscosity of the oil by measuring the diameters of the bubbles produced and their velocity. All tests used a sample with an oil volume of 450 mL. Tests were carried out at different rates of increase of the applied voltage: 0.5, 2, 3 and 5 kV/s. A total of 32 dielectric strength tests were conducted with the capture of the corresponding images. A waiting time of approximately 5 min was established between each cycle to guarantee the disappearance of possible bubbles. In each experiment, the rupture voltage was indicated as well as the number of frames per second (fps) necessary to calculate the velocity of the bubbles produced after the arc occurred. The average capture recording time was 16 s for each experiment, giving an average number of images per experiment of 1344, with a mean time between frames of 0.012 s. As an example, Table 7 shows the results corresponding to the voltage increase rate of 0.5 kV/s. Electric arc at the instant of the breakdown of the electrical strength is shown in Figure 21.

Here, fps is frames per second and R, C and L refer to the right, central and left cameras, respectively. In Figure 4, R, C and L correspond to A, B and C, respectively.

We used a particular experiment to establish the boundary conditions. More specifically, in these simulations, a dielectric breakdown voltage close to 20 kV was set as the boundary condition, as can be seen in Figure 13, Figure 15 and Figure 17. We considered this to be a number consistent with the experimental tests performed. Table 7 shows the dielectric breakdown voltage of a particular set of experiments for a voltage ramp of 0.5 kV/s.

### 5.3. Experimental Measurement of the Dynamic Viscosity of Oil

#### 5.3.1. Theoretical Basis

In [19], an equation is proposed to predict the terminal velocity of the rise of isolated bubbles in Newtonian liquids.

The formulation combines a balance of forces obtained from the limit layer theory for spherical bubbles with an analytical equation from a mechanical energy balance. This is the model denoted model 2 (Mod2). The equation proposed in [19], which gives an approximation of the terminal velocity of the bubble inside the oil, is shown below as Equation (20).
(20)$$V_{T}=\frac{1}{{\left(\frac{1}{V_{T1}^{2}}+\frac{1}{V_{T2}^{2}}\right)}^{0.5}}\;,$$
where $V_{T1}$ is calculated using Equation (21).
(21)$$V_{T1}=V_{Tpot}{\left[1+0.73667\frac{{\left(gD_{i}\right)}^{0.5}}{V_{Tpot}}\right]}^{0.5}$$$V_{Tpot}$ is calculated using Equation (22)
(22)$$V_{Tpot}=\frac{1}{36}\frac{\Delta \rho D_{i}^{2}}{\mu_{oil}}$$
and the term $V_{T2}$ is calculated using Equation (23).
(23)$$V_{T2}=\left[\frac{3\sigma_{b}}{\rho_{oil}}+\frac{gD_{i}\Delta \rho }{2\rho_{oil}}\right].$$

The procedure proposed in this paper consists of solving for the dynamic viscosity $\mu_{oil}$, knowing the terminal velocity of the bubble and its diameter. Using these equations, we start from the experimentally measured terminal velocity $V_{T}$, the diameter of the bubble *D_i_* from the images of the sensors, the difference in gas and oil densities ∆ρ, the acceleration due to gravity g and the viscosity $\mu_{oil}$. The surface tension of the bubble $\sigma_{b}$ is obtained from the Laplace law, as expressed in Equation (24).
(24)$$\begin{array}{cc} \sigma_{b}=\frac{PR}{2}, & P=P_{0}+\rho_{oil}\cdot g\cdot h_{oil}, \end{array}$$
where R=Di2 and *P* is the absolute pressure, which is the atmospheric pressure at a height of 326 m above sea level plus the mean pressure of the oil column.

From these equations, the viscosity is found as a function of the terminal velocity $V_{T}$ obtained from the second-degree algebraic equation given by Equation (25),
(25)$$x^{2}\left(V_{T}^{2}k_{4}k_{2}^{2}-k_{2}^{2}\right)+x\left(V_{T}^{2}k_{4}k_{5}-k_{5}\right)+V_{T}^{2}=0.$$
where x=1μoil is the unknown. The constants $k_{i}$ are as follows.
(26)$$k_{2}=\frac{\Delta \rho D_{i}^{2}}{36},$$
(27)$$k_{3}=0.73667{\left(gD_{i}\right)}^{0.5},$$
(28)$$k_{4}=\frac{1}{V_{T2}^{2}},$$
(29)$$k_{5}=k_{2}\cdot k_{3}.$$

A simpler approximation than Equation (25) is to assume that the acceleration of the bubble is equal to zero. It was found in all the experiments that the terminal velocities of the bubbles were substantially constant. This simplifies the equilibrium equation between the buoyancy force and the Stoke friction force, as indicated in Equation (30).
(30)$$\rho_{oil}\cdot g\frac{4}{3}\pi R^{3}+6\pi R\cdot \mu_{oil}\cdot V_{T}≃0.$$

We also started from $V_{T}$, solving for the dynamic viscosity coefficient from Equation (30). This model was denoted Mod1.

#### 5.3.2. Experimental Results for a Rate of Increase of 0.5 kV/s

Figure 22 represents 14 positions in the movement of a bubble for the experiment corresponding to Table 8. The time between two consecutive positions is equivalent to four frames. The time between frames is 0.0118 s. The image was previously scaled using the ImageJ program [45], which is useful for the subsequent image treatment and better definition of the bubbles. As can be seen in Figure 23, the contour of the bubble is improved, enabling a better measurement of its diameter.

The experimental results corresponding to a dielectric breakdown voltage with a rate of increase of 0.5 kV/s, for a mean dielectric breakdown voltage of 25.725 kV with a standard deviation of ±2.8303 kV and a mean room temperature of 17.2 °C are shown in Table 8.

#### 5.3.3. Comparison of the Kinematic Viscosity with the Proposed Method and Manufacturer’s Data

Figure 24 represents the variation in kinematic viscosity with temperature using the manufacturer’s data as starting data [46]. For this purpose, the following kinematic viscosities were used: 10.5 cSt, 2.7 cSt and 950 cSt, for temperatures of 40 °C, 100 °C and −30 °C, respectively. With these data and starting from the two- and three-parameter models [47], the continuous black and red lines, respectively, were obtained, giving the viscosity values for temperatures from 0 to 100 °C. The discrete data point represented by a circle corresponds to the experimental measurement carried out in our laboratory, as shown in Table 8, by the standard method [39]. The discrete data point represented by a triangle corresponds to the result of applying Equation (30) to the evolution over time of a bubble recorded with the image sensor, according to Table 8. The mean error was less than 0.50%.

## 6. Conclusions

This paper describes an experimental study of the dielectric strength of transformer oil based on the IEC 60156 standard. The contribution made in this paper consists of characterizing the behavior of the oil before and after the electric arc breakdown, by combining a low-cost CMOS imaging sensor and a new matrix of electrical permittivity *M_ε_* associated with the dielectric oil, using the 3D cell method. The root mean square percentage error compared to the finite element method was less than 0.36%.

The IEC 60156 standard test indicates the effective value of the breakdown voltage. However, information on the distribution of Kelvin forces an instant before the dynamic behavior of the arc begins is lost, as is information on the gases that occur an instant after the moment of the electric arc breakdown in the oil. In this paper, after analyzing the images after rupture with a low-cost CMOS imaging sensor, the dynamic viscosity of the oil was indirectly estimated by measuring the rate of rise of the bubbles. These results were compared with a standard method (ASTM D445), and an error of less than 0.5% was obtained.

## 7. Annex

R^2^: coefficient of determination.
(31)$$R^{2}=1-\frac{\sum_{i=1}^{N}{\left(X_{i}-Y_{i}\right)}^{2}}{\sum_{i=1}^{N}{\left(X_{i}-\bar{Y}\right)}^{2}}$$

Value: $-1\leq R^{2}\leq 1$, the nearer to 1, the better.

Advantage: indicates the proximity to the regression line. The perfect regression line has slope of 1.

Disadvantages: it does not always indicate a linear correlation between the data. If the sample is small, the data may, when enlarged, indicate a nonlinear correlation.

Consult: [36].

RMSPE: root mean square percentage error.
(32)$$RMSPE=\left(\sqrt{\frac{1}{N}\overset{N}{\underset{i=1}{\sum }}{\left(X_{i}-Y_{i}\right)}^{2}}\right)\times \frac{1}{\bar{Y}}$$

Value: −1≤RMSPE≤1, the nearer to 0, the better.

Advantage: it is dimensionless and can be used to compare models.

Disadvantages: may underestimate the true measurement as it tries to reproduce the actual data.

Consult: [37].

MAEP: mean absolute percentage error.
(33)$$MAEP=\frac{1}{N}\overset{N}{\underset{i=1}{\sum }}\left|X_{i}-Y_{i}\right|\times \frac{1}{\bar{Y}}$$

Value:−1≤MAEP≤1, the nearer to 0, the better.

Advantage: it is a dimensionless and robust measure of error.

Disadvantages: may underestimate the true measurement as it tries to reproduce the actual data.

Consult: [37].

PBIAS: percent bias.
(34)$$PBIAS=\frac{\sum_{i=1}^{N}\left(X_{i}-Y_{i}\right)}{\sum_{i=1}^{N}Y_{i}}\times 100$$

Value: −1≤PBIAS≤1, the nearer to 0, the better.

Advantage: mean of the deviations, or difference in bias errors, or simply systemic error. It detects aberrant errors or extreme values. If the value is equal to or greater than 1 it is indicative of the existence of these extreme values. Measures the percentage tendency of the simulated data to be larger or smaller than the reference data.

Disadvantages: may underestimate the true measurement as it tries to reproduce the actual data.

Consult: [38].

## Figures and Tables

**Figure 1 sensors-21-07380-f001:**
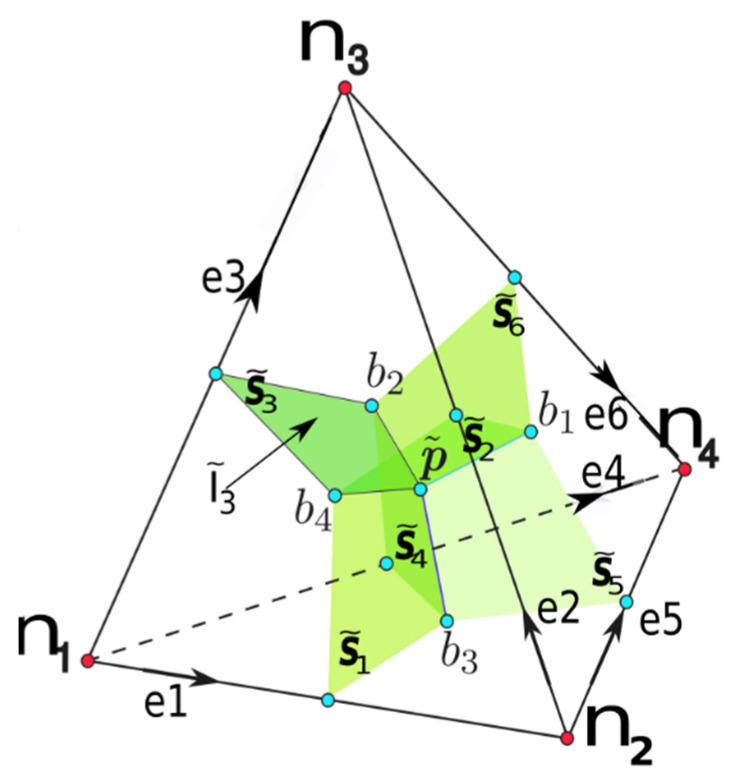
Reference tetrahedron for equations programming. The elements of the primal mesh are represented in black and dual planes in green. A current associated with the dual plane ${\tilde{S}}_{3}$ is also shown.

**Figure 2 sensors-21-07380-f002:**
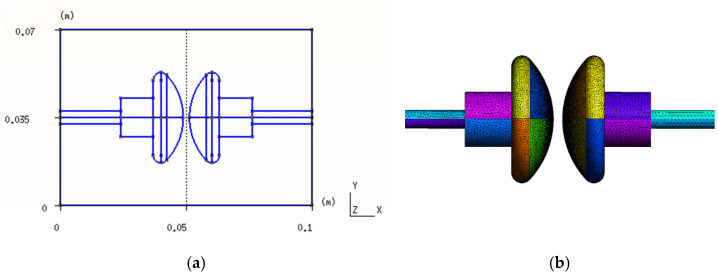
(**a**) Dimensions and coordinates of VDE-type electrodes. (**b**) Surface mesh, with 65,846 triangles, for the electrodes used in the tests.

**Figure 3 sensors-21-07380-f003:**
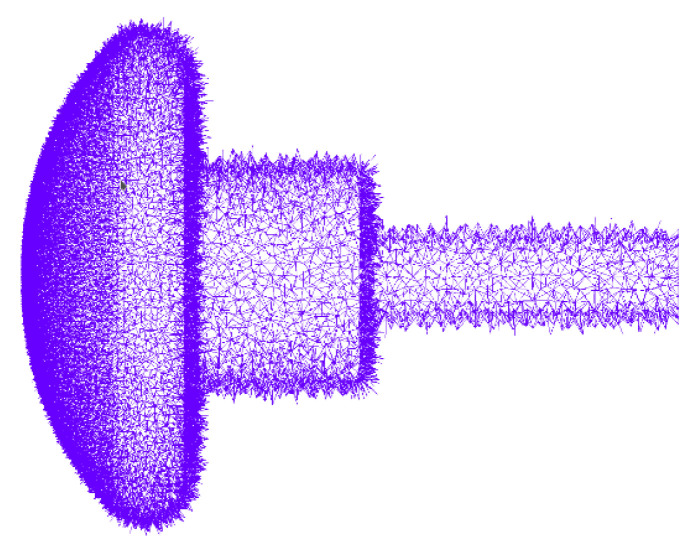
The 51,679 edges corresponding to the relative cohomology between the oil volume and the electrode surface, to obtain the incidence vector *I_c_* from Equation (15).

**Figure 4 sensors-21-07380-f004:**
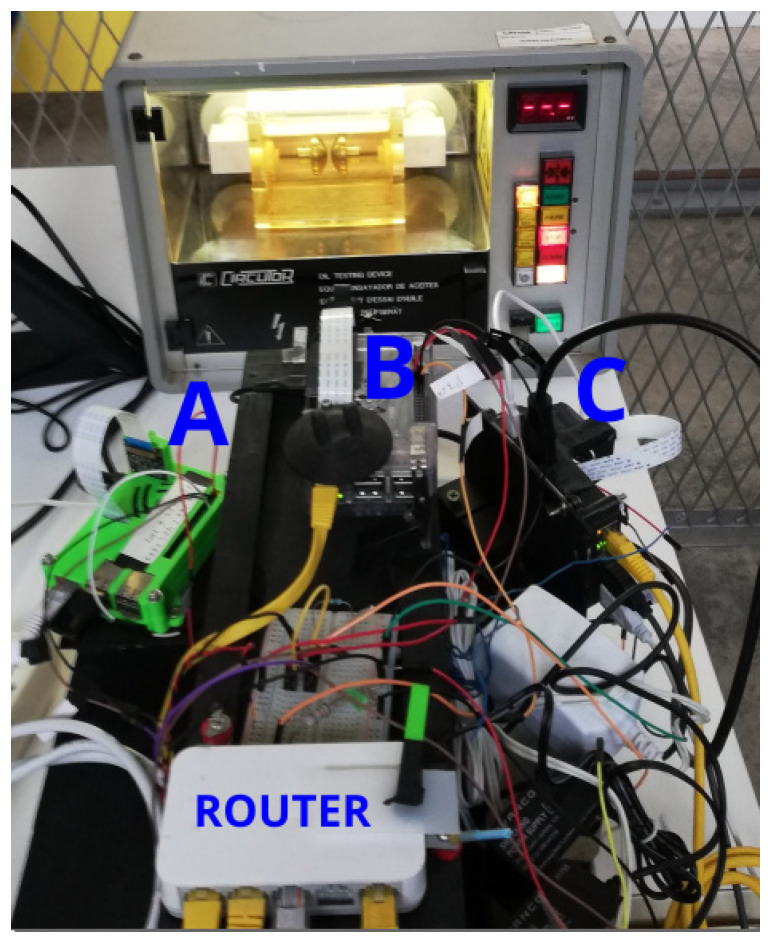
Cameras A, B and C connected with Raspberry Pi and router for the collection of images in a central computer.

**Figure 5 sensors-21-07380-f005:**
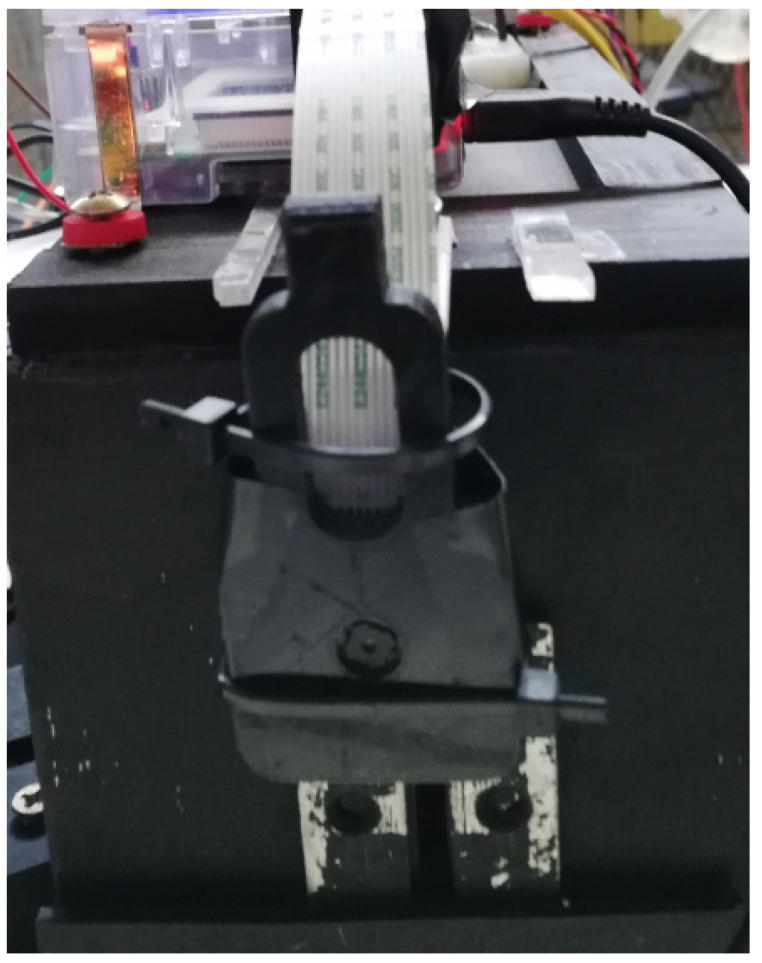
Front view of central camera B.

**Figure 6 sensors-21-07380-f006:**
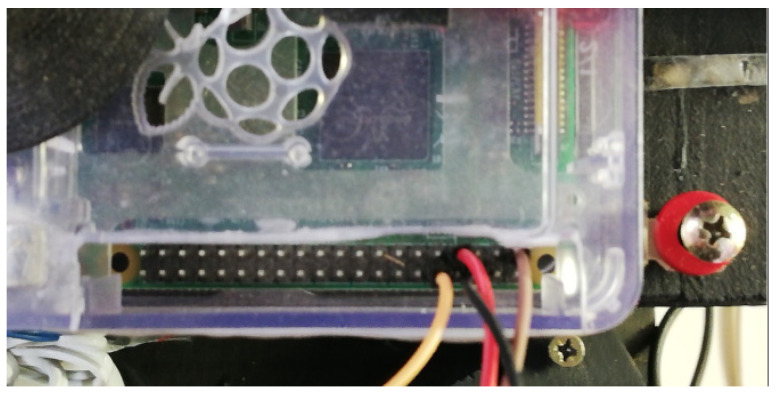
40-pin GPIO connectors to synchronize the triggering of the three cameras.

**Figure 7 sensors-21-07380-f007:**
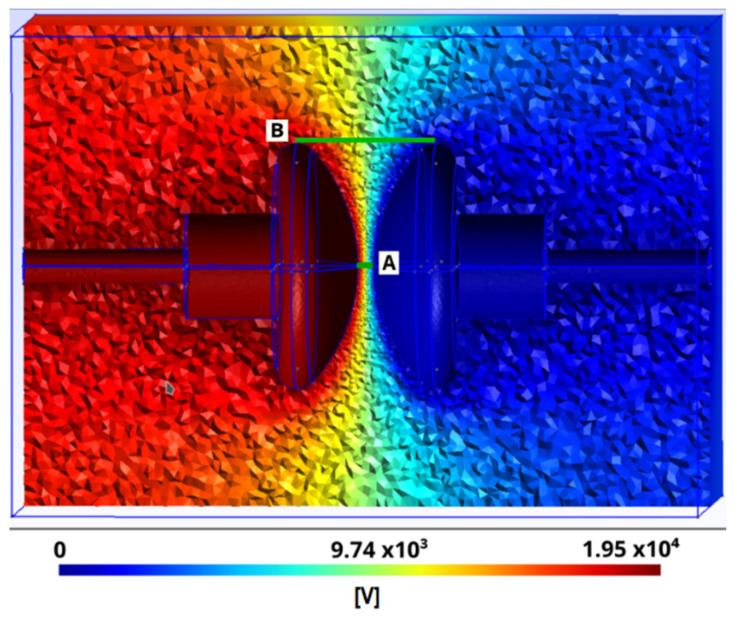
Electrical scalar potential φ obtained in CM when solving system of equations shown in Equation (16), meshed with 2,790,589 tetrahedra and cuts A and B for comparison.

**Figure 8 sensors-21-07380-f008:**
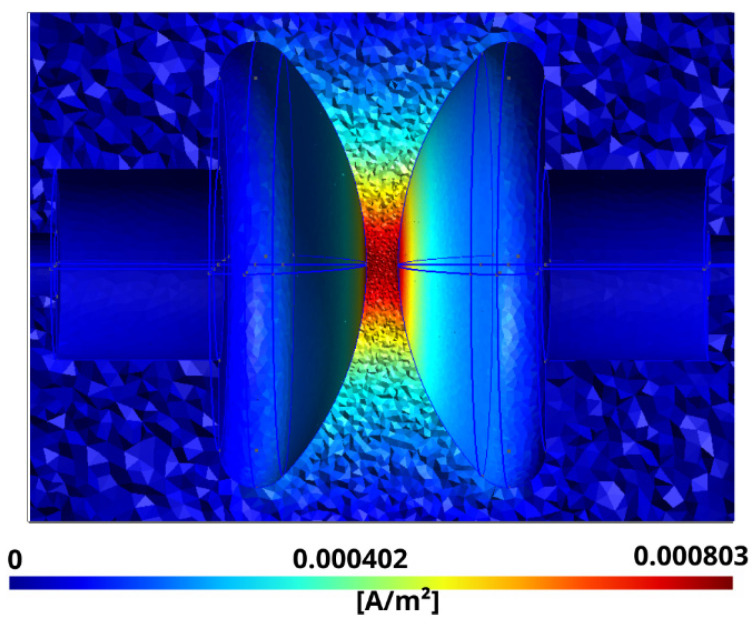
Module of the volumetric density of the conductive current ${\vec{J}}_{f}$ corresponding to Equation (1).

**Figure 9 sensors-21-07380-f009:**
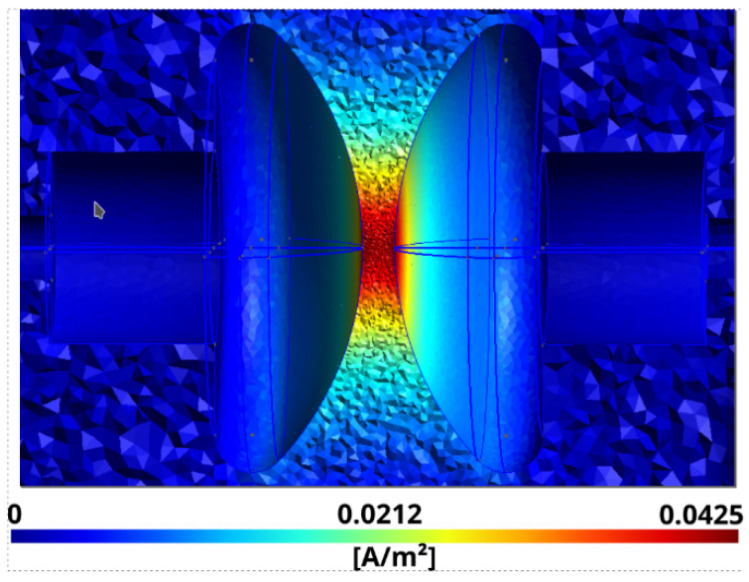
Module of the volumetric density of the displacement current ${\vec{J}}_{d}$ corresponding to Equation (4).

**Figure 10 sensors-21-07380-f010:**
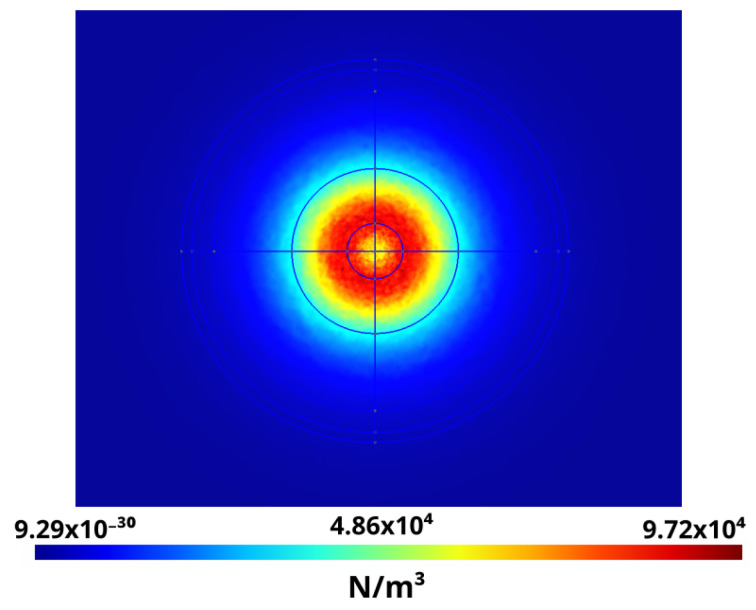
Kelvin force density divided by $\frac{1}{2}\left(\varepsilon -\varepsilon_{oil}\right)$ cut by the plane 1 X + 0 Y +0 Z − 0.0489 = 0.

**Figure 11 sensors-21-07380-f011:**
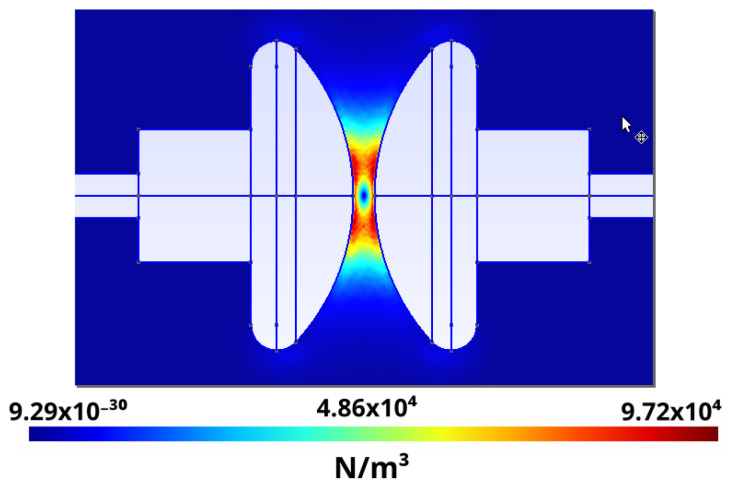
Kelvin force density divided by $\frac{1}{2}\left(\varepsilon -\varepsilon_{oil}\right)$ cut by the plane 0 X + 1 Y +0 Z − 0.035 = 0.

**Figure 12 sensors-21-07380-f012:**
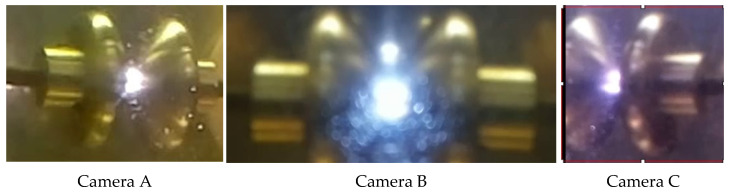
Capture of the images of the electric arc by cameras (**A**–**C**).

**Figure 13 sensors-21-07380-f013:**
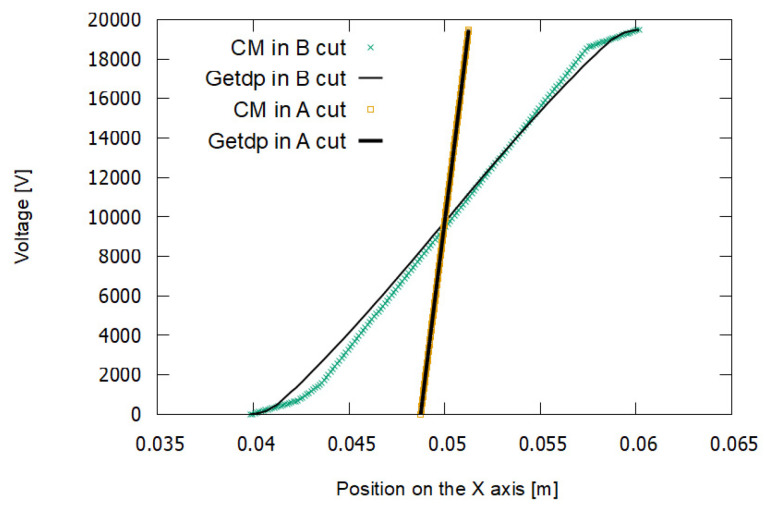
Voltage in cut A (steeper curves) and cut B (curves with lower slope) for experiment E1. The dielectric breakdown voltage is 19,488 V.

**Figure 14 sensors-21-07380-f014:**
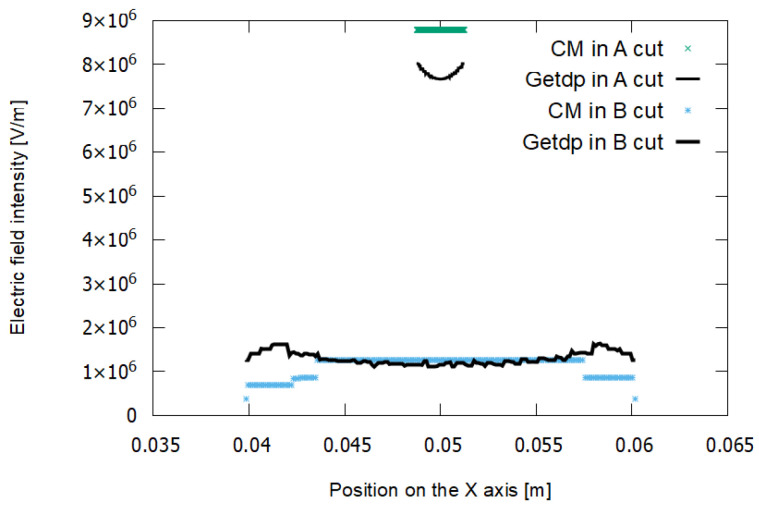
Electric field in cut A (upper curves) and cut B (lower curves) for experiment E1.

**Figure 15 sensors-21-07380-f015:**
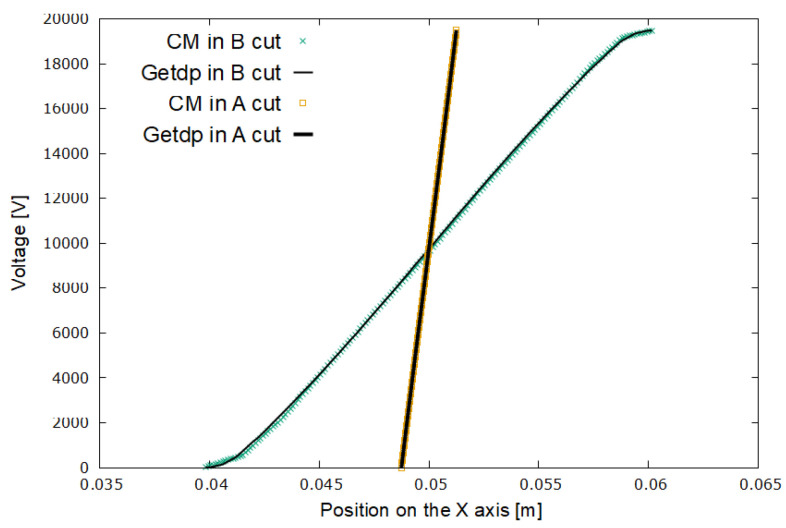
Voltage in cut A (steeper curves) and cut B (curves with lower slope) for experiment E2. The dielectric breakdown voltage is 19,488 V.

**Figure 16 sensors-21-07380-f016:**
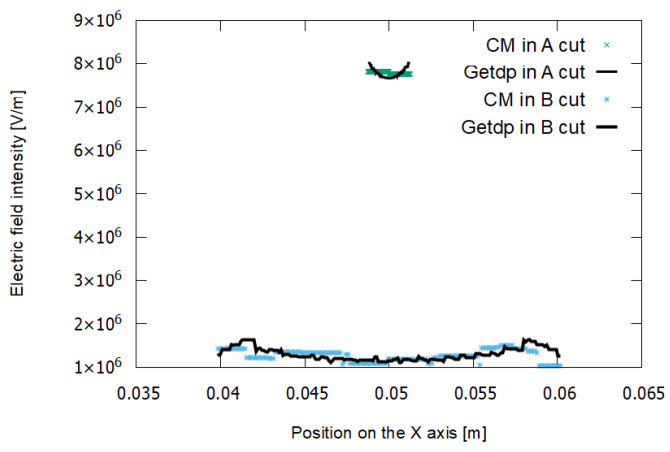
Electric field in cut A (upper curves) and cut B (lower curves) for experiment E2.

**Figure 17 sensors-21-07380-f017:**
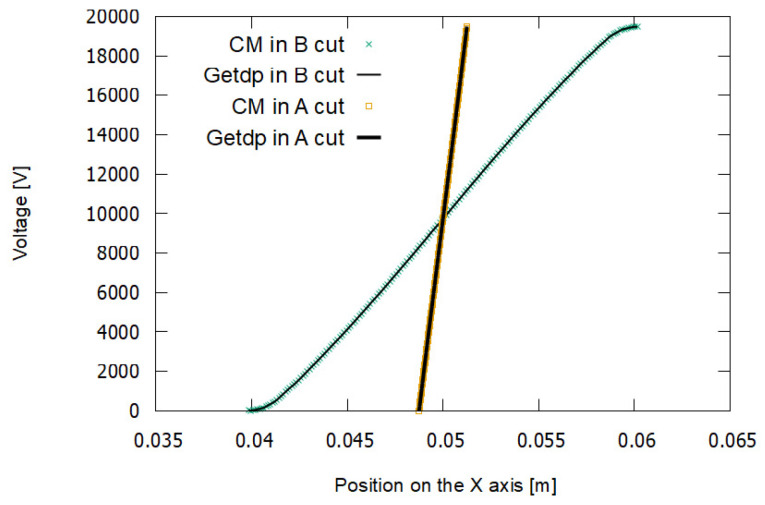
Voltage in cut A (steeper curves) and cut B (curves with lower slope) for experiment E3. The dielectric breakdown voltage is 19,488 V.

**Figure 18 sensors-21-07380-f018:**
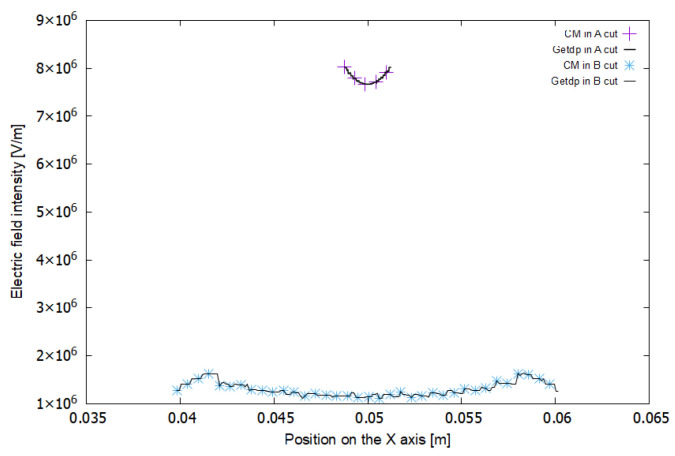
Electric field in cut A (upper curves) and cut B (lower curves) for experiment E3.

**Figure 19 sensors-21-07380-f019:**
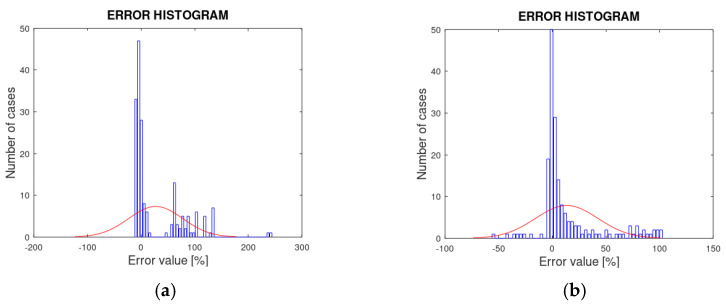
Error histograms: (**a**) comparison C1; (**b**) comparison C4; (**c**) comparison C9; (**d**) comparison C12.

**Figure 20 sensors-21-07380-f020:**
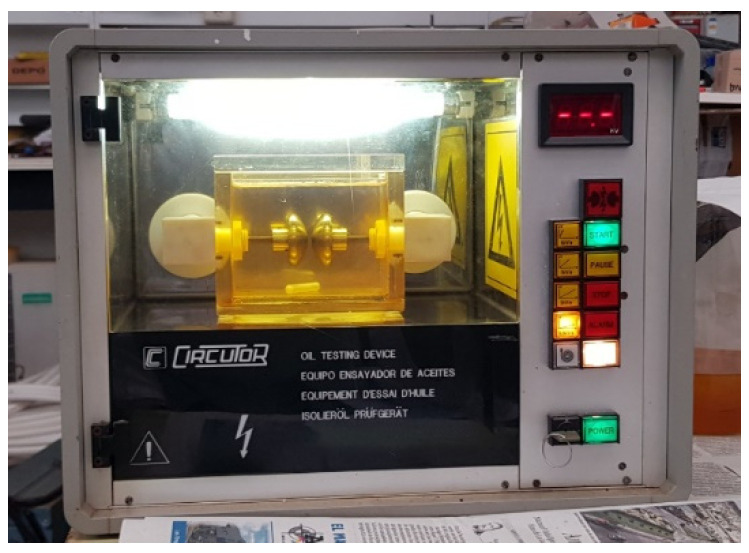
Oil testing device with oil sample.

**Figure 21 sensors-21-07380-f021:**
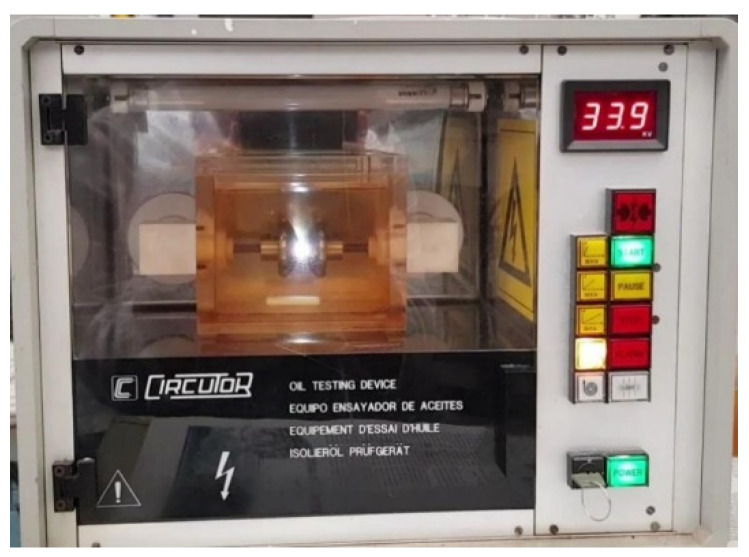
Electric arc at the instant of the breakdown of the electrical strength, with a value of the dielectric breakdown voltage of 33.9 kV.

**Figure 22 sensors-21-07380-f022:**
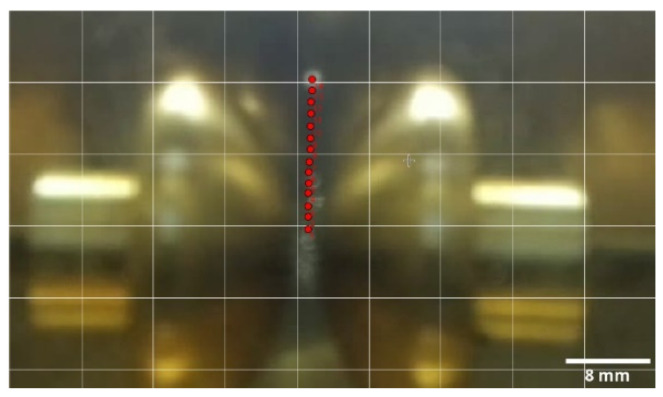
Position of the bubble at 14 instants of time for the experiment in Table 8.

**Figure 23 sensors-21-07380-f023:**
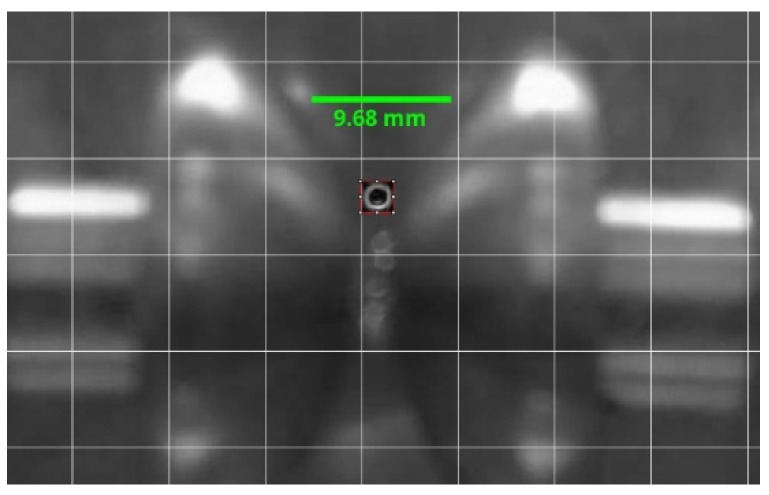
Improvement in the definition of the bubble using the ImageJ program for measuring the diameter.

**Figure 24 sensors-21-07380-f024:**
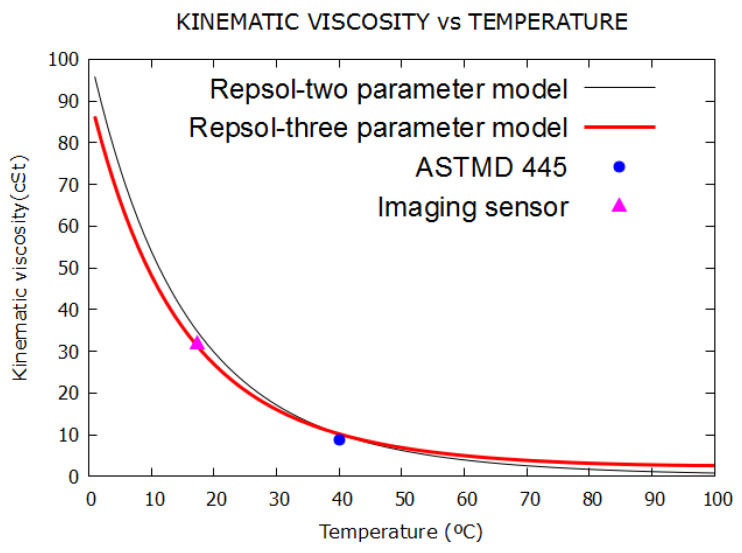
Kinematic viscosity versus temperature in dielectric oil.

**Table 1 sensors-21-07380-t001:** Mesh characteristics and physical properties considered in the numerical experiments.

Experiment	Nodes	Edges	Faces	Volumes	Conductivity [S/m]	Relative Permittivity	BDV [V]
**E1**	1175	6467	9687	4394	10^−^^10^	3.8	19,488
**E2**	5889	34,737	54,142	25,293	10^−^^10^	3.8	19,488
**E3**	487,435	3,362,385	5,665,540	2,790,589	10^−^^10^	3.8	19,488

**Table 2 sensors-21-07380-t002:** Comparisons of developed numerical experiments.

C1	Numerical experiment E1-Cut A. Electric field.
C2	Numerical experiment E1-Cut B. Electric field.
C3	Numerical experiment E1-Cut A. Voltage.
C4	Numerical experiment E1-Cut B. Voltage.
C5	Numerical experiment E2-Cut A. Electric field.
C6	Numerical experiment E2-Cut B. Electric field.
C7	Numerical experiment E2-Cut A. Voltage.
C8	Numerical experiment E2-Cut B. Voltage.
C9	Numerical experiment E3-Cut A. Electric field.
C10	Numerical experiment E3-Cut B. Electric field.
C11	Numerical experiment E3-Cut A. Voltage.
C12	Numerical experiment E3-Cut B. Voltage.

**Table 3 sensors-21-07380-t003:** Metrics of the proposed comparisons.

Comparison	C1	C2	C3	C4	C5	C6	References
R^2^ [0, +1] Optimum: +1	0.5772	0.0000	1.0000	0.9979	0.0000	0.1083	[36]
RMSPE [−1, +1] Optimum: 0	0.3624	0.1127	0.0045	0.0512	0.0152	0.1302	[37]
MAEP [−1, +1] Optimum: 0	0.2409	0.1119	0.0040	0.0405	0.0129	0.0959	[37]
PBIAS [−1, +1] Optimum: 0	0.1447	−0.1260	0.0000	0.0216	0.0001	0.0254	[38]

**Table 4 sensors-21-07380-t004:** Metrics of the proposed comparisons.

Comparison	C7	C8	C9	C10	C11	C12	References
R^2^ [0, +1] Optimum: +1	1.0000	0.9998	0.9997	1.0000	1.0000	1.0000	[36]
RMSPE [−1, +1] Optimum: 0	0.0050	0.0101	0.0003	0.0001	0.0000	0.0000	[37]
MAEP [−1, +1] Optimum: 0	0.0044	0.0086	0.0002	0.0000	0.0000	0.0231	[37]
PBIAS [−1, +1] Optimum: 0	−0.0019	0.0054	0.0000	0.0000	0.0000	0.0000	[38]

**Table 5 sensors-21-07380-t005:** Technical characteristics.

OT-40	Value
Supply voltage	230 V
Test voltage	0–41 kV
Consumption	100 VA/800 VA
Frequency	50 Hz
Measurement tolerance	±2%
Response time on disconnection	<20 ms
Test standards	UNE EN 60156:1997
Work temperature	+15/+25 °C
Exterior size	385 × 300 × 400 mm
Weight	38 kg

**Table 6 sensors-21-07380-t006:** Results of the oil sample tests.

Parameters	Unit	Method	Value
Kinematic viscosity at 40 °C	cSt	ASTM D445	8.75
Density at 15 °C	g/mL	ASTM D4052	0.858
Flashpoint	°C	ASTM D92	145.0
Water content	ppm	ASTM D1533	20.40
Color	-	ASTM D1500	1.0
Total acidity	mg KOH/g	IEC 61125C	0.06

**Table 7 sensors-21-07380-t007:** Breakdown voltage. Ramp of 0.5 kV/s.

Sample Number	fps			T [°C]	V_DBV_ [kV]	Agitator [Y/N]
	R	C	L			
1	87.0	84.7	86.2	17.1	24.4	Y
2	87.0	84.7	86.2	17.1	24.4	Y
3	87.0	84.7	86.3	17.1	23.5	Y
4	86.9	84.7	86.2	17.2	23.3	Y
5	87.0	85.0	86.2	17.3	23.1	Y
6	87.0	84.7	86.1	17.3	29.1	Y
7	87.0	84.7	86.1	17.3	30.0	Y
8	87.1	84.7	86.1	17.3	28.0	Y

**Table 8 sensors-21-07380-t008:** Determination of the kinematic viscosity of the oil using gas bubbles.

Experiment No.	Bubble Diameter [mm]	Bubble Mean Velocity [m/s]	Voltage [kV]	Viscosity (cSt) Mod1	Viscosity (cSt) Mod2
1	1.33	0.028	24.4	33.6	54.7
2	1.34	0.030	24.4	32.1	49.8
3	1.28	0.026	23.5	34.6	60.4
4	1.33	0.028	23.1	33.2	54.3
5	1.34	0.030	29.1	31.9	49.3
6	1.26	0.032	30.0	26.9	39.0
			Mean 25.75	Mean 32.05	Mean 51.25

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
