# Peer review of "Characterization of Dielectric Oil with a Low-Cost CMOS Imaging Sensor and a New Electric Permittivity Matrix Using the 3D Cell Method"

_sensors, 2021, doi:10.3390/s21217380_

Round 1

Reviewer 1 Report

The work is interesting. Quantitative result analysis and critical discussion are expected. Further improvement is required as list below.

  1. More research background including more transient electromagnetic measurement and imaging with J. reference are required e.g. Liang Ge, etc., Study on a new electromagnetic flowmeter based on three-value trapezoidal wave excitation, Flow Measurement and Instrumentation, Volume 78, 2021,101882,ISSN 0955-5986, https://doi.org/10.1016/j.flowmeasinst.2020.101882.
  2. Figures 13-18 need more quantitative analysis and comparison;
  3. It would be good to highlight advantages and disadvantages of the proposed work. 

Author Response

Reviewer 1

The manuscript has undergone an extensive English revision and has been checked by a native English-speaking colleague.

The work is interesting. Quantitative result analysis and critical discussion are expected.

Further improvement is required as list below.

  • More research background including more transient electromagnetic measurement and imaging with J. reference are required e.g. Liang Ge, etc., Study on a new electromagnetic flowmeter based on three-value trapezoidal wave excitation, Flow Measurement and Instrumentation, Volume 78, 2021,101882,ISSN 0955-5986, https://doi.org/10.1016/j.flowmeasinst.2020.101882.

Answer:

Thank you very much for your comments.

We have improved the research background and included six more electromagnetic references in the state of the art, as can be seen from the new text included in the Introduction section, as follows:

“In CM, the topological equations obtained directly from Maxwell's laws are exact — balance equations — while the constitutive equations — obtained from the discretization process — are approximate. In the latter case, source-type quantities defined in the elements of the dual mesh must be related to the configuration quantities corresponding to the elements of the primal mesh [10]. Field magnitudes and physical properties of the medium are assumed constant at least in the primal mesh. This ensures that the discrete equations are consistent with the continuous constitutive equations, in the sense that the discrete constitutive equations approximate the continuous constitutive equations with an error that decreases with the mesh size [11].

Most of the research papers in CM literature focus on the construction of discrete constitutive equations. Among those related to a quasi-electrostatic problem in 2D, with plane symmetry, is [12]. In this work, an isotropic and anisotropic electrostatic field is studied by means of CM. In [13], the electrostatic problem is studied in 2D with plane symmetry. The constitutive equation is used with two approximations. The first approach assumes a uniform field with triangular base inside each primal cell, and the second approach, more general, assumes the uniformity of the fields in subregions of each primal cell with quadrilateral base. In [14], a 2D analysis with axial symmetry (axisymmetric) is made for a quasi-electrostatic problem for a gas insulated line for the ITER Neutral Beam Injector.

In [15], an electrostatic induction micro-motor, using the CM in 2D, is studied.

In the literature, to the best of the authors’ knowledge, the 3D cell method has not been used to simulate dielectric breakdown voltage tests on transformer oils in a quasi-electrostatic problem. In this article, it is proposed to use for the electrical permittivity constitutive matrix a geometric structure analogous to the matrix that appears in [16] for an electromagnetic problem in 3D to calculate eddy currents.

The advantage of using the same geometric structure in the constitutive matrices — of conductivity and permittivity — in the quasi-electrostatic problem in 3D is that it reduces the complexity of the programmed source code and the execution times. This is so because the constitutive matrix is calculated in the assembly of the system of equations only once for each tetrahedron. The electrical conductivity and electrical permittivity properties of each tetrahedron are multiplied by the common matrix. This is done element by element, until the final system of equations is completed.”

We have also added to the research state of the art a comment about different proposed standards used for the determination of the dielectric breakdown voltage of insulating liquids.

“There are three main standards for the determination of the dielectric breakdown voltage of insulating liquids: ASTM D1816 - 12(2019), ASTM D877 / D877M – 19 and IEC 60156:2018. In this paper, UNE EN 60156, which is based on IEC 60156:2018, was followed.”

As can be seen in Equation (16), we work in sinusoidal steady-state regimen.

  • Figures 13-18 need more quantitative analysis and comparison;

Answer:

In the section “4.1 Validation of the numerical simulations”, the following comment were added:

In the E1 experiment, a low-density mesh is used. In Figure 13 it can be seen that the voltage values in FEM-getdp and CM are almost coincident for cut A. However, they differ more for cut B. In Figure 14 there is a notable difference between the electric values of the electric field given by FEM-getdp and those of CM both in cut A and cut B because the mesh is not very dense.

In experiment E2, a medium-density mesh is used. In Figure 15 it can be seen that the voltage values in FEM-getdp and CM are practically the same. In Figure 16 there are still differences between the electric field values given by FEM-getdp and CM both in cut A and cut B.

A high-density mesh is used in experiment E3. In Figures 17 and 18 there are no differences for the values given by FEM-getdp and CM for both voltage and electric field.

Therefore, it is observed that as the mesh density is increased the two methods turn out to give coincident solutions. This confirms the validity of the new matrix proposed in this article for CM.

  • It would be good to highlight advantages and disadvantages of the proposed work.

Answer:

In CM [7-16], the physical laws governing the physical phenomena are expressed in integral formulation. The final algebraic equation system is tailored directly without discretization of the differential equations. The fact that final algebraic equation system is tailored directly without discretization of the differential equations is an important advantage. In addition, as we say in the Introduction section, “Using this methodology, the analysis of the electric fields created around the electric arc immersed in dielectric oil greatly facilitates the establishment of the conditions of contour and continuity when working with global magnitudes and directly develops the system of equations without the need to discretize the differential equations”.

The advantage of using the same geometric structure in the constitutive matrices — of conductivity and permittivity — in the quasi-electrostatic problem in 3D is that it reduces the complexity of the programmed source code and the execution times. This is so because the constitutive matrix is calculated in the assembly of the system of equations only once for each tetrahedron. The electrical conductivity and electrical permittivity properties of each tetrahedron are multiplied by the common matrix. This is done element by element, until the final system of equations is completed.

Another advantage, as is said in the Abstract, is that a new method is proposed to measure the kinematic viscosity of dielectric oils using a low-cost image sensor.

Reviewer 2 Report

The authors developed a considerable work, since extensive numerical and experimental tests were conducted. In my view, however, the paper has not been well designed. The authors do not motivate the need of using the CM, which in my view is equivalent to FEM for the considered application. In particular, the electric constitutive relationship (with matrix Mε) does not seems to be used together with experimental analysis, as claimed in the Abstract. Section V appears to be completely uncorrelated with the rest of the paper since it describes an experimental methodology for assessing (non-electrical) physical properties of transformer oils. 

There are also some other concerns about the scientific soundness of the proposed methodology: 

  • The model geometry shown in Fig. 2 is axisymmetric and can be solved by a standard 2-D FEM software. Why was a full 3D analysis used? What are the reasons for a lack of symmetry which calls for 3D analysis?
  • The equation system (16) has to be solved under suitable boundary conditions, which have not been indicated. For instance, are you imposing the breakdown voltage at electrodes in Fig. 2?  
  • If voltage has been imposed, the total current can be computed at the post-processing stage from the electric potential distribution. The total current is not needed for computing the polarization force. Note that this quantity has been defined twice, both in (5) and (15). 
  • It is well known that CM for elliptic-problem provides the same stiffness matrix of 1st order FEM. Therefore, the same results have to be obtained. Note that the mass matrix, defined in (1), cannot be found in literature. I have found only a short remark in the Appendix here: Lorenzo Codecasa, Francesco Trevisan, Constitutive equations for discrete electromagnetic problems over polyhedral grids, Journal of Computational Physics, Volume 225, Issue 2, 2007, Pages 1894-1918, https://doi.org/10.1016/j.jcp.2007.02.032.
  • There is no quantitative comparison between numerical results obtained from CM and the experimental results. Figs. 7 - 11 show only some color maps of different outputs from the CM code. 
  • In Section 4.2 stochastic error analysis has been used for validating the correctness of a deterministic method, i.e. the Cell Method. I do not see actually any source of randomness in data, which motivates the use of stochastic analysis. This happens, for instance, when the variance of equation coefficients is accounted for in the model.

Author Response

Reviewer 2

We have carried out a thorough revision of the manuscript and have corrected the English grammar and style. The manuscript has been checked by a native English-speaking colleague.

The authors developed a considerable work, since extensive numerical and experimental tests were conducted.

Answer:

Thank you very much.

In my view, however, the paper has not been well designed. The authors do not motivate the need of using the CM, which in my view is equivalent to FEM for the considered application.

Answer:

We concur with Reviewer 2 that CM is equivalent to FEM for the considered application. However, in CM [7-16], the physical laws governing the physical phenomena are expressed in integral formulation. The final algebraic equation system is tailored directly without discretization of the differential equations. The fact that final algebraic equation system is tailored directly without discretization of the differential equations is an important advantage. In addition, as we say in the Introduction section, “Using this methodology, the analysis of the electric fields created around the electric arc immersed in dielectric oil greatly facilitates the establishment of the conditions of contour and continuity when working with global magnitudes and directly develops the system of equations without the need to discretize the differential equations.”.

The advantage of using the same geometric structure in the constitutive matrices - of conductivity and permittivity - in the quasi-electrostatic problem in 3D is that it reduces the complexity of the programmed source code and the execution times. This is so because the constitutive matrix is calculated in the assembly of the system of equations only once for each tetrahedron. The electrical conductivity and electrical permittivity properties of each tetrahedron are multiplied by the common matrix. This is done element by element, until the final system of equations is completed.

In particular, the electric constitutive relationship (with matrix Mε) does not seem to be used together with experimental analysis, as claimed in the Abstract.

Answer:

You are right. The abstract was not clear on this point and has now been clarified.

There are two main objectives in the paper. One is to present a new matrix of electrical permittivity, , that predicts the electric field before and after the electric rupture is reached. The other one is to measure the kinematic viscosity of the dielectric oil through a low-cost CMOS imaging sensor that measures the distribution of bubbles, their diameters and their rates of ascent after the electric arc takes place. In addition, we have also presented the experiments that were performed to estimate the dielectric breakdown voltage in order to obtain the boundary conditions for CM and FEM simulations. In this way, both objectives are related.

Section V appears to be completely uncorrelated with the rest of the paper since it describes an experimental methodology for assessing (non-electrical) physical properties of transformer oils.

Answer:

In this section, the breakdown voltage of the oil is obtained, which is necessary to establish the boundary conditions of the numerical method that is developed in section 4. In section V, other physical properties such as kinematic viscosity are also analyzed.

There are also some other concerns about the scientific soundness of the proposed methodology:

  • The model geometry shown in Fig. 2 is axisymmetric and can be solved by a standard 2-D FEM software. Why was a full 3D analysis used? What are the reasons for a lack of symmetry which calls for 3D analysis?

Answer:

The electrodes have axisymmetric gometry, but the oil container is cubic, and this makes the global model non-axisymmetric, and so we have to perform a 3D analysis. Furthermore, the new proposed electrical permittivity matrix is a 3D matrix and with these calculations generality to the matrix Mε is given.

  • The equation system (16) has to be solved under suitable boundary conditions, which have not been indicated. For instance, are you imposing the breakdown voltage at electrodes in Fig. 2?

Answer:

You are right, the boundary conditions of the system of equations Equation (16) consist of the imposition of the dielectric breakdown voltage at electrodes in Figure 2.

  • If voltage has been imposed, the total current can be computed at the post-processing stage from the electric potential distribution. The total current is not needed for computing the polarization force. Note that this quantity has been defined twice, both in (5) and (15).

Answer:

This is true because the voltage has been imposed through the dielectric breakdown voltage. It is also true that the total current is not needed for computing the polarization force but the total current mentioned in Equation (5) and Equation (15) is included in the system of equations Equation (16). This current is necessary to establish an equivalent circuit of the electrodes immersed in oil. Knowledge of this current will be used in future works in which the tangent delta test will be applied to the insulating oil.

  • It is well known that CM for elliptic-problem provides the same stiffness matrix of 1st order FEM. Therefore, the same results have to be obtained.

Answer:

You are right. Thank you.

  • Note that the mass matrix, defined in (1), cannot be found in literature. I have found only a short remark in the Appendix here: Lorenzo Codecasa, Francesco Trevisan, Constitutive equations for discrete electromagnetic problems over polyhedral grids, Journal of Computational Physics, Volume 225, Issue 2, 2007, Pages 1894-1918, https://doi.org/10.1016/j.jcp.2007.02.032.

Answer:

Thank you very much for the new reference that you have given us and that we have now included among the references in the revised version of the manuscript.

In the literature, as far as the authors are aware, the 3D CM has not been used to simulate dielectric breakdown voltage tests on transformer oils in a quasi-electrostatic problem. In this article, it is proposed to use for the electrical permittivity constitutive matrix a geometric structure analogous to the matrix that appears in [16] for a 3D electromagnetic problem to calculate the eddy currents.

  • There is no quantitative comparison between numerical results obtained from CM and the experimental results. Figs. 7 - 11 show only some color maps of different outputs from the CM code.

Answer:

You are right. Results shown in Figures 7-11 are color maps only for CM because they look exactly the same for FEM. The detailed comparison between the two methods is explained in section 4.1, “Validation of the numerical simulations”, where we compare CM and FEM.

  • In Section 4.2 stochastic error analysis has been used for validating the correctness of a deterministic method, i.e., the Cell Method. I do not see actually any source of randomness in data, which motivates the use of stochastic analysis. This happens, for instance, when the variance of equation coefficients is accounted for in the model.

Answer:

Metrics are not stochastic problem-solving methods. Metrics are applied to the validation of a model against a reference or pattern. In this case, the model to be validated are the results obtained with CM and the reference or pattern are the results obtained with FEM. CM and FEM are approximate numerical methods, therefore not exact. In both methods a tolerable error is pre-set.

There are various sources of error. One is the truncation of the figures and the accumulation of errors due to the numerical operations performed. Another, which is the one that most affects our problem, is the layout of the cuts. The proximity of the cuts to the nodes of the mesh makes the calculated value at the cut more accurate. In contrast, when the cut moves further away from the node, it will be necessary to obtain interpolated values, thus producing a greater error. This happens in both FEM and CM.

In this work, an analysis was carried out in FEM with a very dense mesh. These results were used as a reference. Different mesh densities were established in CM. With the results obtained in CM, different comparisons were made with the results of the high mesh density FEM model. Various statistical indicators (metrics) were used to check the validity of CM versus FEM.

Table 2 shows all the numerical simulations carried out. Table 3 shows the statistics used, as well as their optimal values. Based on the theory of errors, for a comparison between two perfectly identical models, the value of the error should be zero. A theoretical model of error distribution has a normal distribution of errors, with an error mean around zero, as can be seen in Figure 19 in red. As can be seen in Figure 19, in blue, in this case they are approximate numerical methods and, therefore, there are errors, the mean of which is around zero.

Therefore, Section 4.2 demonstrates the validity of CM versus FEM as an approximate numerical method.

To clarify the interpretation of metrics, we have introduced an appendix at the end of the paper introducing the formulas of the metrics used in Table 3.

Reviewer 3 Report

Minor changes / typos:

Remove the "full stop" in the title, please

In the abstract, please try to avoid the use of "we". Instead, us impersonal form.

Line 25: after and before -> before and after
Line 25: The error made -> The made error

Line 40: among others. [1, 2]. -> among others [1, 2].

Line 51: joule effect -> Joule effect

Line 78: we propose -> it is proposed

Line 98: consists in -> consists of

Line 122: the results obtained -> the obtained results 

Line 148: , see figure 1. -> , as shown in Figure 1.
The same in Line 152
Check this format mistake in all the document (Lines 169, 176, 184, 216, 225, 287, 294, 306, 358, 364, 426, 438...)

Line 151: vectors ,correspond -> vectors, correspond

Line 161: on the insulating material used -> on the used insulating material

Line 171: Equation (3) is repetead

Line 217: carge -> charge

Line 259: they can be seen in figure 4 -> as shown in Figure 4
and a detail of the central camera in figure 5. -> and a detail of the central camera in Figure 5.Line 283: has -> and has 

Line 365: in table 1 -> in Table 1

Line 382: Figure 15 and Figure 16 -> Figures 15 and 16

Line 404: table 2. -> Table 2.

Line 409: Table 3 and table 4 -> Tables 3 and 4

Line 416: of the comparisons proposed -> of the proposed comparisons

Line 423: numerical experiments analyzed. -> analyzed numerical experiments.

Line 451: Experiments performed -> Performed experiments

Line 467: results obtained -> obtained results  

Line 534: temperatura -> temperature

General Comments:Introduction and state of the art should be improve to highlight the scientific contributions of the paper.

Author Response

Reviewer 3

We have undertaken a thorough revision of the manuscript and have corrected the English grammar and style. The manuscript has been checked by a native English-speaking colleague.

Minor changes / typos:

Answer:

Thank you very much for your detailed suggestions.

General Comments:

Introduction and state of the art should be improved to highlight the scientific contributions of the paper.

Answer:

We have improved the research state of the art and have commented on different proposed methods in different standards.

We have included in the Introduction section the following:

“In CM, the topological equations obtained directly from Maxwell's laws are exact — balance equations — while the constitutive equations — obtained from the discretization process — are approximate. In the latter case, source-type quantities defined in the elements of the dual mesh must be related to the configuration quantities corresponding to the elements of the primal mesh [10]. Field magnitudes and physical properties of the medium are assumed constant at least in the primal mesh. This ensures that the discrete equations are consistent with the continuous constitutive equations, in the sense that the discrete constitutive equations approximate the continuous constitutive equations with an error that decreases with the mesh size [11].

Most of the research papers in CM literature focus on the construction of discrete constitutive equations. Among those related to a quasi-electrostatic problem in 2D, with plane symmetry, is [12]. In this work, an isotropic and anisotropic electrostatic field is studied by means of CM. In [13], the electrostatic problem is studied in 2D with plane symmetry. The constitutive equation is used with two approximations. The first approach assumes a uniform field with triangular base inside each primal cell, and the second approach, more general, assumes the uniformity of the fields in subregions of each primal cell with quadrilateral base. In [14], a 2D analysis with axial symmetry (axisymmetric) is made for a quasi-electrostatic problem for a gas insulated line for the ITER Neutral Beam Injector.

In [15], an electrostatic induction micro-motor, using the CM in 2D, is studied.

In the literature, to the best of the authors’ knowledge, the 3D cell method has not been used to simulate dielectric breakdown voltage tests on transformer oils in a quasi-electrostatic problem. In this article, it is proposed to use for the electrical permittivity constitutive matrix a geometric structure analogous to the matrix that appears in [16] for an electromagnetic problem in 3D to calculate eddy currents.

The advantage of using the same geometric structure in the constitutive matrices — of conductivity and permittivity — in the quasi-electrostatic problem in 3D is that it reduces the complexity of the programmed source code and the execution times. This is so because the constitutive matrix is calculated in the assembly of the system of equations only once for each tetrahedron. The electrical conductivity and electrical permittivity properties of each tetrahedron are multiplied by the common matrix. This is done element by element, until the final system of equations is completed.”

We have also added to the research state of the art a comment about different proposed standards used for the determination of the dielectric breakdown voltage of insulating liquids.

We have also added to the research state of the art a comment about different proposed standards used for the determination of the dielectric breakdown voltage of insulating liquids.

“There are three main standards for the determination of the dielectric breakdown voltage of insulating liquids: ASTM D1816 - 12(2019), ASTM D877 / D877M – 19 and IEC 60156:2018. In this paper UNE EN 60156, which is based on IEC 60156:2018, was followed.”

The contributions of the paper include the fact that knowing the behavior of the oil an instant after the dielectric breakdown voltage allows us to also know, by means of a low-cost imaging sensor in an experimental test, the distribution of velocities and diameters of the bubbles released by the gases and, on this basis, to calculate the kinematic viscosity of the oil.

The usefulness and applicability of the calculation of the kinematic viscosity can be seen in the comment made in the Introduction section, “oils help to evacuate the heat generated due to hysteresis losses and eddy currents in iron, as well as the losses due to the Joule effect in the transformer coils. This last condition requires a low coefficient of dynamic viscosity of the oil”.

However, the behavior of the oil before the dielectric breakdown voltage is obtained through CM numerical simulations using the new matrix that we have proposed in this article. This gives us the distribution of the Kelvin forces an instant before the dielectric breakdown voltage.

The usefulness of calculating Kelvin forces an instant before the dielectric breakdown voltage helps to understand where the dielectric breakdown voltage occurs. Figures 10 and 11 show that maximum forces are in a concentric ring around the electrode but not in the middle of it. This can be of interest for electrode designers and is coherent with the experimental images shown in Figure 12 obtained with imaging sensors.

Reviewer 4 Report

This paper proposes a method to characterize dielectric-oil using a CMOS Imaging Sensor and a new electric permittivity matrix.

I suggest a revision based on the following points in order to improve the quality of the paper and to find a more suitable journal where to send it:

1.- English grammar and style must be deeply revised all along the manuscript.

2.- Please consider if is better “dielectric breakdown voltage” instead of “dielectric rigidity”.

3.- Abstract. Please rephrase the following paragraph because it is unclear: “In this standard the effective value of the breakdown voltage is collected, but it does not provide information on the distribution of Kelvin forces an instant before the dynamic behavior of the arc begins and monitoring of the state of the gases that are produced an instant after the moment of breaking of the electric arc in the oil.”

4.- Page 3. “Our contribution consists in characterizing the behavior of the oil an instant, before and after the electric arc rupture”. It is not clear the interest or applicability of this characterization. Please, develop since it is a key point to prove the usefulness and applicability of this research work.

5.- The Introduction Section lacks of a “research state of the art”, as well as similar methods proposed in different standards: NEMA, ISO, EN, etc.

6.- Please describe the meaning of CM the first time it appears in the text

7.- Breakdown is usually anticipated by a rise of the current. The authors must explain how the software decides when the breakdown condition is attained. Is there any threshold value? How is it settled?

8.- In Fig. 18, in cut A there is no trace corresponding to the cell method. Please clarify this point.

9.- The formulas of metrics in Table 3 must be included in the text.

  1. Calculated and measured values of the breakdown voltage are very different. Please explain these differences.

11.- Conclusions need to be redone.

I believe the remarks above will help to improve the paper.

Author Response

Reviewer 4

This paper proposes a method to characterize dielectric-oil using a CMOS Imaging Sensor and a new electric permittivity matrix.

I suggest a revision based on the following points in order to improve the quality of the paper and to find a more suitable journal where to send it:

1.- English grammar and style must be deeply revised all along the manuscript.

Answer:

Thank you very much. We have undertaken a thorough review of the manuscript and have corrected the English grammar and style. The manuscript has been checked by a native English-speaking colleague.

2.- Please consider if is better “dielectric breakdown voltage” instead of “dielectric rigidity”.

Answer:

We agree. It is better to say, “dielectric breakdown voltage”, and have now included this in the nomenclature.

3.- Abstract. Please rephrase the following paragraph because it is unclear: “In this standard the effective value of the breakdown voltage is collected, but it does not provide information on the distribution of Kelvin forces an instant before the dynamic behavior of the arc begins and monitoring of the state of the gases that are produced an instant after the moment of breaking of the electric arc in the oil.”

Answer:

Thank you very much. We think it can be improved. We have introduced a few modifications in this paragraph of the Abstract.

“In the standardized test only the effective value of the dielectric breakdown voltage is collected. However, the information on the distribution of Kelvin forces [18] is lost an instant before the dynamic behavior of the arc begins, as is information on the gases that are produced an instant after the moment of breaking of the electric arc in the oil.”

4.- Page 3. “Our contribution consists in characterizing the behavior of the oil an instant, before and after the electric arc rupture”. It is not clear the interest or applicability of this characterization. Please, develop since it is a key point to prove the usefulness and applicability of this research work.

Answer:

Knowing the behavior of the oil an instant after the dielectric breakdown voltage, it is also possible to know, by means of a low-cost imaging sensor in an experimental test, the distribution of velocities and diameters of the bubbles released by the gases and, on this basis, to calculate the kinematic viscosity of the oil.

The usefulness and applicability of the calculation of the kinematic viscosity can be seen in the comment made in the Introduction section, “oils help to evacuate the heat generated due to hysteresis losses and eddy currents in iron, as well as the losses due to the Joule effect in the transformer coils. This last condition requires a low coefficient of dynamic viscosity of the oil”.

However, the behavior of the oil before the dielectric breakdown voltage is obtained through CM numerical simulations using the new matrix that we have proposed in this article. This gives us the distribution of the Kelvin forces an instant before the dielectric breakdown voltage.

The usefulness of calculating Kelvin forces an instant before the dielectric breakdown voltage helps to understand where the dielectric breakdown voltage occurs. Figures 10 and 11 show that maximum forces are located in a concentric ring around the electrode but not in the middle of it. This can be of interest for electrode designers and is coherent with the experimental images shown in Figure 12 obtained with imaging sensors.

5.- The Introduction Section lacks of a “research state of the art”, as well as similar methods proposed in different standards: NEMA, ISO, EN, etc.

Answer:

We have improved the research state of the art and have commented on different proposed methods in different standards.

We have included in the Introduction section the following:

“In CM, the topological equations obtained directly from Maxwell's laws are exact — balance equations — while the constitutive equations — obtained from the discretization process — are approximate. In the latter case, source-type quantities defined in the elements of the dual mesh must be related to the configuration quantities corresponding to the elements of the primal mesh [10]. Field magnitudes and physical properties of the medium are assumed constant at least in the primal mesh. This ensures that the discrete equations are consistent with the continuous constitutive equations, in the sense that the discrete constitutive equations approximate the continuous constitutive equations with an error that decreases with the mesh size [11].

Most of the research papers in CM literature focus on the construction of discrete constitutive equations. Among those related to a quasi-electrostatic problem in 2D, with plane symmetry, is [12]. In this work, an isotropic and anisotropic electrostatic field is studied by means of CM. In [13], the electrostatic problem is studied in 2D with plane symmetry. The constitutive equation is used with two approximations. The first approach assumes a uniform field with triangular base inside each primal cell, and the second approach, more general, assumes the uniformity of the fields in subregions of each primal cell with quadrilateral base. In [14], a 2D analysis with axial symmetry (axisymmetric) is made for a quasi-electrostatic problem for a gas insulated line for the ITER Neutral Beam Injector.

In [15], an electrostatic induction micro-motor, using the CM in 2D, is studied.

In the literature, to the best of the authors’ knowledge, the 3D cell method has not been used to simulate dielectric breakdown voltage tests on transformer oils in a quasi-electrostatic problem. In this article, it is proposed to use for the electrical permittivity constitutive matrix a geometric structure analogous to the matrix that appears in [16] for an electromagnetic problem in 3D to calculate eddy currents.

The advantage of using the same geometric structure in the constitutive matrices — of conductivity and permittivity — in the quasi-electrostatic problem in 3D is that it reduces the complexity of the programmed source code and the execution times. This is so because the constitutive matrix is calculated in the assembly of the system of equations only once for each tetrahedron. The electrical conductivity and electrical permittivity properties of each tetrahedron are multiplied by the common matrix. This is done element by element, until the final system of equations is completed.”

We have also added to the research state of the art a comment about different proposed standards used for the determination of the dielectric breakdown voltage of insulating liquids.

“There are three main standards for the determination of the dielectric breakdown voltage of insulating liquids: ASTM D1816 - 12(2019), ASTM D877 / D877M – 19 and IEC 60156:2018. In this paper UNE EN 60156, which is based on IEC 60156:2018, was followed.”

6.- Please describe the meaning of CM the first time it appears in the text.

Answer:

Thank you. We have described the meaning of CM the first time it appears.

7.- Breakdown is usually anticipated by a rise of the current. The authors must explain how the software decides when the breakdown condition is attained. Is there any threshold value? How is it settled?

Answer:

The testing device Circutor OT-40 cannot measure currents. It is an analog device and has no software. It determines when the dielectric breakdown voltage is produced and its magnitude through analogical systems.

8.- In Fig. 18, in cut A there is no trace corresponding to the cell method. Please clarify this point.

Answer:

Thank you very much. We have changed Figure 18 introducing the trace corresponding to the cell method. We have also renamed the vertical axis of Figures 13 to 18 because there was as a mistake the original version.

9.- The formulas of metrics in Table 3 must be included in the text.

Answer:

Thank you. We have introduced an appendix at the end of the paper introducing the formulas of the metrics used in Table 3.

10.- Calculated and measured values of the breakdown voltage are very different. Please explain these differences.

Answer:

The article contains various CM numerical simulations that always keep the boundary conditions constant, exclusively modifying the mesh density to determine its convergence. These results allow to verify the new proposed CM matrix, comparing it with FEM analysis performed with a very fine mesh. More specifically, in these simulations, a dielectric breakdown voltage close to 20 kV was set as the boundary condition, as can be seen in Figures 13, 15 and 17. We think it is a round number coherent with the experimental tests that we have realized. Table 7 shows the dielectric breakdown voltage of a particular set of experiments for a voltage ramp of 0.5 kV/s.

11.- Conclusions need to be redone.

Answer:

We have revised and improved the conclusions of the paper and we think we have clarified them.

“This paper shows an experimental study of the dielectric strength of transformer oil based on the IEC 60156 standard. The contribution made in this paper consists of characterizing the behavior of the oil before and after the electric arc break, combining a low-cost CMOS imaging sensor and a new matrix of electrical permittivity M_ε associated with the dielectric oil using the 3D cell method. The root mean square percentage error compared to the finite element method is less than 0.36%.

The standardized test IEC 60156 indicates the effective value of the breakdown voltage. However, information on the distribution of Kelvin forces is lost an instant before the dynamic behavior of the arc begins, as is information on the gases that occur an instant after the moment of the electric arc breaking in the oil. In this paper, after analyzing the images after rupture with a low-cost CMOS imaging sensor, the dynamic viscosity of the oil was indirectly estimated by measuring the rate of rise of the bubbles. These results were compared with a standard method (ASTM D445) and an error of less than 0.5% was obtained.”

I believe the remarks above will help to improve the paper.

Answer:

Thank you very much.

Round 2

Reviewer 2 Report

The authors unfortunately do not show the actions taken during the review. In reading the revised version of the manuscript I cannot see any improvement in several important parts.

Just to summarize the main issues still present in the new manuscript: 

  • The total current is yet defined twice by eqn. (5) and (15)
  • Boundary conditions are not mentioned again in Section 2.3. There is only a short sentence in the Introduction at line 154. Matrix system (13) does not make much sense, since the first row is uncoupled from the second one. The total current is simply a post-processing quantity to be computed at the post-processing. 
  • There is no comparison with experiments. Of course color maps of FEM and CM are the same because these methods lead to the same system matrices. 

There are some meaningless formulas. See, e.g.,  

  • Line 259 "U=-GφU"
  • Line 280 "rot x E = 0"

Please note that there are still several spelling errors (e.g. "reacheing" line 35, page 1). I would suggest to use the error checker in Word. 

Author Response

REVIEWER-2

The authors unfortunately do not show the actions taken during the review. In reading the revised version of the manuscript I cannot see any improvement in several important parts.

Just to summarize the main issues still present in the new manuscript: 

  • The total current is yet defined twice by eqn. (5) and (15).

Answer:

You are right. The total current has been defined twice by Equation (5) and (15) because Equation (5) shows the two main components of the total current – the displacement current and the conductive electric current –, while Equation (15) details its explicit composition based on its physical and geometric properties.

To clarify this point, we have added the following text on line 272.

“Note that the total current has been defined twice by Equation (5) and (15) because Equation (5) shows the two main components of the total current – the displacement current and the conductive electric current – while Equation (15) details its explicit composition based on its physical and geometric properties.”

  • Boundary conditions are not mentioned again in Section 2.3. There is only a short sentence in the Introduction at line 154. Matrix system (13) does not make much sense, since the first row is uncoupled from the second one. The total current is simply a post-processing quantity to be computed at the post-processing. 

Answer:

  1. a) Boundary conditions are not mentioned again in Section 2.3. There is only a short sentence in the Introduction at line 154.

You are right. Boundary conditions are not mentioned in Section 2.3.  Boundary conditions are stabilized on the electrode surfaces. All the nodes on one electrode surface have a value of electric potential equal to zero and all the nodes on the surface of the other electrode have the dielectric breakdown voltage.

Answer:

  1. b) Matrix system (13) does not make much sense, since the first row is uncoupled from the second one. The total current is simply a post-processing quantity to be computed at the post-processing. 

Answer:

We understand that you are talking about system equation (16).

You right. In matrix system (16) the first row is uncoupled from the second one. But we have preferred a compact equation system that includes the total intensity of the electrode which avoids an additional post processing calculus. It is also true that we increase the dimension of the system in one degree of freedom that corresponds to the total intensity through the electrode. In this way, solving a single matrix system all the unknowns – degrees of freedom – are obtained at the same time, without post-processing.

To clarify the document, we have added the following paragraphs on line 276:

 “In matrix system (16) the first row is uncoupled from the second one. But we have preferred a compact equation system that includes the total intensity of the electrode. This avoids an additional post processing calculus. It is also true that we increase the dimension of the system in one degree of freedom that corresponds to the total intensity through the electrode. In this way, solving a single matrix system all the unknowns – degrees of freedom – are obtained at the same time, without post-processing”.

“Boundary conditions are stabilized on the electrode surfaces. All the nodes on one electrode surface have a value of electric potential equal to zero and all the nodes on the surface of the other electrode have the dielectric breakdown voltage.”

  • There is no comparison with experiments. Of course color maps of FEM and CM are the same because these methods lead to the same system matrices. 

Answer:

You are right. There is no comparison of analytical results with experiments. We have presented some experiments that have been done to estimate the dielectric breakdown voltage in order to obtain the boundary conditions for the CM and FEM simulations, that are presented in section 2.3. In this way both objectives are related. We have used a particular experiment to stablish the boundary conditions.

More specifically, in these simulations, a dielectric breakdown voltage close to 20 kV was set as the boundary condition, as can be seen in Figures 13, 15 and 17. We think it is a round number coherent with the experimental tests that we have realized. Table 7 shows the dielectric breakdown voltage of a particular set of experiments for a voltage ramp of 0.5 kV/s.

To clarify this point, we have included the following paragraph in section 5.2:

“We have used a particular experiment to stablish the boundary conditions. More specifically, in these simulations, a dielectric breakdown voltage close to 20 kV was set as the boundary condition, as can be seen in Figures 13, 15 and 17. We think it is a round number coherent with the experimental tests that we have realized. Table 7 shows the dielectric breakdown voltage of a particular set of experiments for a voltage ramp of 0.5 kV/s.”

Besides, the usefulness of calculating Kelvin forces an instant before the dielectric breakdown voltage through CM helps to understand where the dielectric breakdown voltage occurs. Figures 10 and 11 show that maximum forces are located in a concentric ring around the electrode but not in the middle of it. This is coherent with the experimental images shown in Figure 12 obtained with the imaging sensors.

To clarify this point, we have included the following paragraph in section 4.

“Besides, the usefulness of calculating Kelvin forces an instant before the dielectric breakdown voltage, through CM, helps to understand where the dielectric breakdown voltage occurs. Figures 10 and 11 show that maximum forces are located in a concentric ring around the electrode, but not in the middle of it. This is coherent with the experimental images shown in Figure 12, obtained with the imaging sensors.”.

There are some meaningless formulas. See, e.g.,  

Answer:

  • Line 259 "U=-GφU"

Thank you very much, there was a mistake and we have corrected it. "U=-Gφ"

  • Line 280 "rot x E = 0"

You are right. Thank you. It there was a mistake.

We have changed "" by "NABLA x (E) ≈ 0" --- ()

Please note that there are still several spelling errors (e.g. "reacheing" line 35, page 1). I would suggest to use the error checker in Word. 

Answer:

Thank you very much. The document has been checked and minor spelling mistakes have been corrected.

Reviewer 4 Report

The paper can be accepted since the authors have replied all my questions.

Please replace "electric field intensity" by "electric field strength" all along the manuscript

Author Response

REVIEWER-4

Answer:

The document has been checked and minor spelling mistakes have been corrected.

The paper can be accepted since the authors have replied all my questions.

Answer:

Thank you very much.

Please replace "electric field intensity" by "electric field strength" all along the manuscript.

Answer:

Thank you. We have replaced "electric field intensity" by "electric field strength" all along the manuscript.
